# Real-space imaging of a phenyl group migration reaction on metal surfaces

Zilin Ruan [1,3], Baijin Li[1,3], Jianchen Lu [1,3] ✉, Lei Gao [2,3] ✉, Shijie Sun[1], Yong Zhang[1] & Jinming Cai [1] ✉

The explorations to extend present chemical synthetic methods are of great importance to simplify synthetic routes of chemical species. Additionally, understanding the chemical reaction mechanisms is critical to achieve controllable synthesis for applications. Here, we report the on-surface visualization and identification of a phenyl group migration reaction of 1,4-dimethyl-2,3,5,6-tetraphenyl benzene (DMTPB) precursor on Au(111), Cu(111) and Ag(110) substrates. With the combination of bond-resolved scanning tunneling microscopy (BR-STM), noncontact atomic force microscopy (nc-AFM) and density functional theory (DFT) calculations, the phenyl group migration reaction of DMTPB precursor is observed, forming various polycyclic aromatic hydrocarbons on the substrates. DFT calculations reveal that the multiple-step migrations are facilitated by the hydrogen radical attack, inducing cleavage of phenyl groups and subsequent rearomatization of the intermediates. This study provides insights into complex surface reaction mechanisms at the single molecule level, which may guide the design of chemical species.

Conventional chemical synthesis is mainly based on in-solution chemistry, where the precise control of the products depends on the chemical nature of the precursors and is boosted with the presence of catalytic agents. However, the synthetic routes are often hindered by side reactions due to the complex in-solution environment. As a complementary method, on-surface synthesis is an effective approach to fabricate nanostructures with atomic precision. Furthermore, the nanostructures at different on-surface reaction stages can be explored via nanotechnologies, such as scanning probe microscopy[1–8]. So far, strategies employed in on-surface reactions are mostly based on mimicking known in-solution reactions[9–11]. Although some recent on-surface progresses reported different synthetic manners compared to in-solution chemistry[12–14], a more comprehensive study of on-surface reactions to extend the present synthetic protocol as well as the reaction mechanism is highly desirable.

The rearrangement reaction represents a unique class of organic transformations in chemistry[15–23]. Among these, radical aryl migration is of particular interest to the chemical community owing to its potential applications in radical chemistry and organic synthesis[24–28]. Radical aryl migration is the intramolecular radical migration of an aryl group from a carbon or heteroatom to a radical center by forming spirocyclic intermediate or transition state. This migration can be typically terminated by radical pathways such as elimination, β-scission, or H abstraction[24]. The mechanism of radical aryl migration has been long debated, and a large effort has been dedicated in the search for a unequivocal mechanism. These mechanistic studies are also important to elucidate possible synthetic pathways, based on combining radical aryl migrations with different radical reactions, which might in principle allow to synthesize compounds that are difficult to obtain with more traditional nucleophilic rearrangements or other protocols[28–36]. Therefore, the mechanistic studies of these migration reactions, together with their potential synthetic applications, have become an attractive research field in both radical chemistry and organic synthesis over the past decades.

In this article, we demonstrate the on-surface generations of various polycyclic aromatic hydrocarbons via a phenyl group

[1]Faculty of Materials Science and Engineering, Kunming University of Science and Technology, 650093 Kunming, Yunnan, People's Republic of China. [2]Faculty of Science, Kunming University of Science and Technology, 650500 Kunming, Yunnan, People's Republic of China. [3]These authors contributed equally: Zilin Ruan, Baijin Li, Jianchen Lu, Lei Gao. ✉e-mail: jclu@kust.edu.cn; lgao@kust.edu.cn; j.cai@kust.edu.cn

migration reaction on Au(111), Cu(111), and Ag(110) substrates, respectively, which cannot be accessed by in-solution chemistry. The precursor, 1,4-dimethyl-2,3,5,6-tetraphenyl benzene (DMTPB), consisting of a central benzene ring with its six hydrogen atoms substituted by two para-methyl groups and four phenyl groups, undergoes a phenyl migration and cascaded cyclodehydrogenations after thermal treatment. As illustrated in Fig. 1, after annealing of the self-assembled DMTPB clusters at elevated temperature, conventional C−C couplings between phenyl groups in the early stage of the reaction are surprisingly absent (no-go products **N1** and **N2**). Instead, four benzene rings unexpectedly migrate to the para-methyl positions, forming Thiele's hydrocarbon, which further transforms into fully conjugated polycyclic aromatic hydrocarbons **A1**, **A2**, **A3**, **A4**, and **A5** via intramolecular cyclodehydrogenation reactions. Employing the state-of-art bond-resolved scanning tunneling microscopy (BR-STM) and noncontact atomic force microscopy (nc-AFM) techniques, we successfully identified the molecular structures of on-surface reaction products and directly captured the cascaded structural arrangements. The phenyl group migration reaction is furtherly evaluated by density functional theory (DFT) calculations, which show good agreement with experimental observations.

## Results and discussion

DMTPB precursors **1** were deposited via thermal sublimation at 360 K onto Au(111) surface, which was held at room temperature (RT). Subsequent STM imaging reveals regular linear self-assembly (Fig. 2a and Supplementary Fig. 1b), exhibiting alternate bright protrusions. Such features are more pronounced in constant-height STM topography (Supplementary Fig. 1c) and can be assigned to two up-tilted phenyl groups on each side of one methyl group (Supplementary Figs. 1 and 2). The corresponding chemical model of the molecular chain is shown in Supplementary Fig. 1d, in which the benzene rings colored gray-blue denote the up-tilted phenyl groups with respect to the Au(111) surface. The line profile measured across the tilted phenyl rings shows good agreement with the optimized chemical model (Supplementary Fig. 1e), which collaborates with the assignment. The relative position of DMTPB monomer in the self-assembly suggested that the self-

assembled structure is stabilized by π−π and −CH··· π interactions between individual molecules, as illustrated previously[37,38].

We then slowly increased the temperature to 440 K and annealed the sample to trigger the intramolecular reaction. The ordered self-assembly transformed into individual nonplanar monomers with a very high yield (>95%) for the major product (Supplementary Fig. 3). Further annealing to 590 K induces the planarization, and individual species as well as coupled molecular clusters were formed, as shown in Fig. 2b. Careful inspection of the constant-current STM topographies reveals five main products (labeled as **A1** to **A5**) with different yields (see Supplementary Tab. 1), as listed in Fig. 2c–g. The constant-current STM topographies of these products exhibit rather blur microscopic contrasts arising from the electronic local density of state (LDOS) associated with molecule-orbitals (Fig. 2c–g). By contrast, the constant-height BR-STM images (Fig. 2h–l) with the benefit of a CO-functionalized tip unambiguously reveal the backbone of products **A1** to **A5** at the single atomic bond level, whereby the corresponding molecular structures are identified (Fig. 2m−q, see also Supplementary Fig. 4). Interestingly, surface-induced chirality is also observed for products **A2**, **A3**, and **A5** (Supplementary Fig. 5). Based on the structures resolved by the BR-STM imaging, a comparison between the as-synthesized products **A1**−**A5** and the precursors indicates rather a complex structural transformation process, as illustrated in Fig. 1. The expected direct C-C couplings between benzene rings or benzene ring and a methyl group are absent after annealing, while the phenyl groups migrate to the methyl groups instead, resulting in the formation of Thiele's hydrocarbon (referred as **1d**) (Fig. 1) and subsequent cyclo-dehydrogenation isomer products of **A1**−**A5** with analogous electronic properties (Supplementary Fig. 6). This hypothesis is supported by a careful inspection of the chemical structures of the major products obtained at 440 K (Supplementary Figs. 7−9) and the cascaded cyclo-dehydrogenation reaction at higher temperatures up to 590 K (Supplementary Figs. 10 and 11), as well as some rarely observed products (Supplementary Figs. 12 and 13), which are elucidated and discussed in detail in the Supplementary Information (Supplementary Figs. 7−8 and 14-17). In fact, we did not observe the four phenyl groups migrated product **1d**, while a C−C coupling between two migrated phenyl groups of product **1d** was experimentally observed (Supplementary Figs. 7 and 8, product **2**), suggesting the lower reaction barrier from **1d** to **2** due to its high reactive diradical character (Supplementary Fig. 8). It's also noticed that product **A1** with a high yield at 590 K cannot be obtained directly from the dominating intermediate **2** at 440 K, which suggested a possible ring opening of intermediate **2** at higher temperatures (see Supplementary Tab. 1 and its related discussion for more details). The proposed migration mechanism on Au(111) surface is based on the followings: (i) the carbon atoms in the para-methyl group (methyl-C) are connected to the benzene core, where the original phenyl groups are lost; (ii) the carbon atoms remain in the observed cascaded cyclodehydrogenated products at a higher temperature, which suggests a structural transformation rather than the decomposition of the precursor; (iii) the cascaded cyclodehydrogenations are directly captured by our step-by-step thermal annealing, which agrees well with the possible structural evolution of intermediate 1d; (iv) the cleavages of phenyl rings have also been observed, which is similar to the beginning situation of the migration reaction[39–41]; (v) previous discussion on the role of methyl groups in the early stage of thermal treatment of polycyclic aromatic hydrocarbons indicates that it might lead to the migration of functional groups[42].

To probe the feasibility of these proposed reaction pathways, we then performed DFT calculations of reaction pathways for elementary steps from precursor **1** to experimentally observed product **2**. Figure 3a shows a detailed step-by-step migration of phenyl groups migration. The reaction mechanism from the precursor **1** to **1a** is proposed as follows: The H radicals are easily produced from the initial

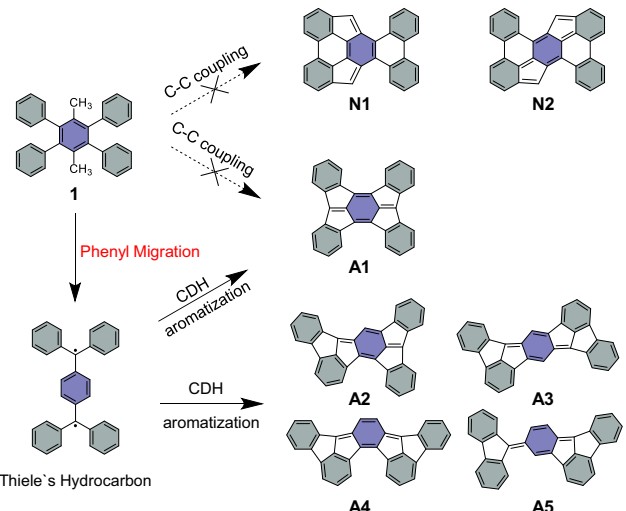

**Fig. 1 | On-surface generation of polycyclic aromatic hydrocarbons A1, A2, A3, A4, and A5 via a phenyl group migration reaction in this work.** The direct cyclodehydrogenations (CDH) between phenyl groups or phenyl group and methyl groups are absent here (no-go products **N1** and **N2**). The phenyl groups migrate to the para-methyl groups and subsequent cyclodehydrogenation produces **A1**, **A2**, **A3**, **A4**, and **A5**. Shaded rings with two different colors mark the four phenyl groups and the central benzene, respectively.

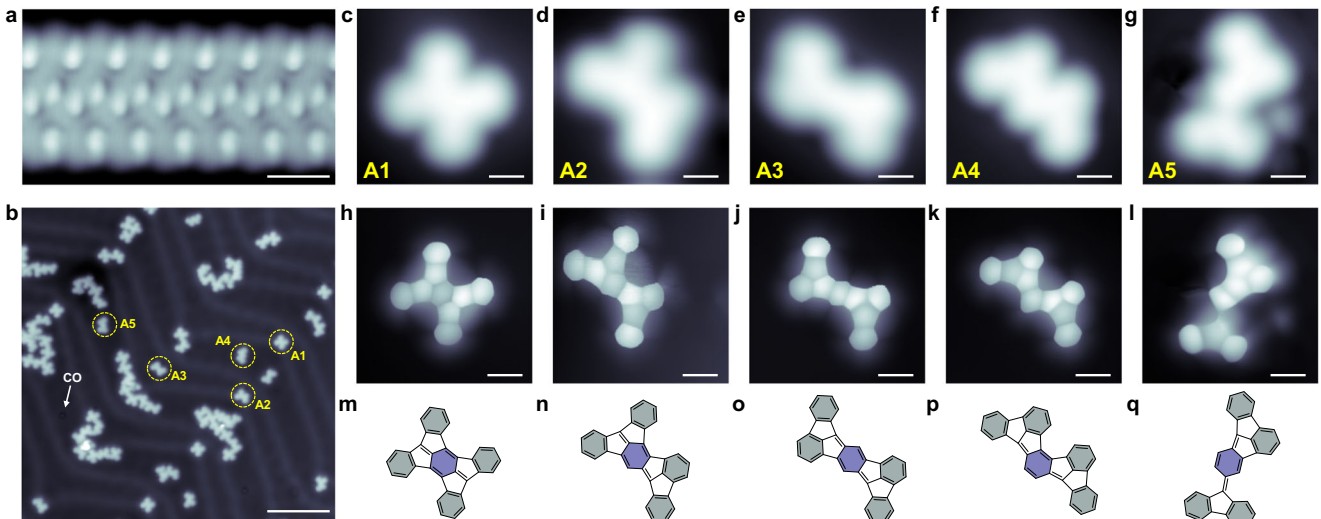

**Fig. 2 | Products formed after thermal annealing of DMTPB on Au(111) at 590 K.**
**a** STM topographies of linear self-assembly of DMTPB on Au(111). **b** Large-scale STM topography (CO tip, $I = 50$ pA, $V_s = 100$ mV) after thermal annealing at 590 K. **c–g** Constant current STM topographies (CO tip, $I = 50$ pA, $V_s = 100$ mV) of the products **A1**, **A2**, **A3**, **A4**, and **A5**, respectively, as denoted in (**b**) with dashed yellow circles. **h–l** Constant height BR-STM images (CO tip, $V_s = 2$ mV) of the products corresponding to (**c–g**), respectively. **m–q** The identified molecular structures of five monomer products corresponding to **A1**, **A2**, **A3**, **A4**, and **A5** in panels **h–l**. Shaded rings with two different colors mark the four phenyl groups and the central benzene in the precursor. Scale bar: **a** 1 nm; **b** 7 nm; **c–l** 0.5 nm.

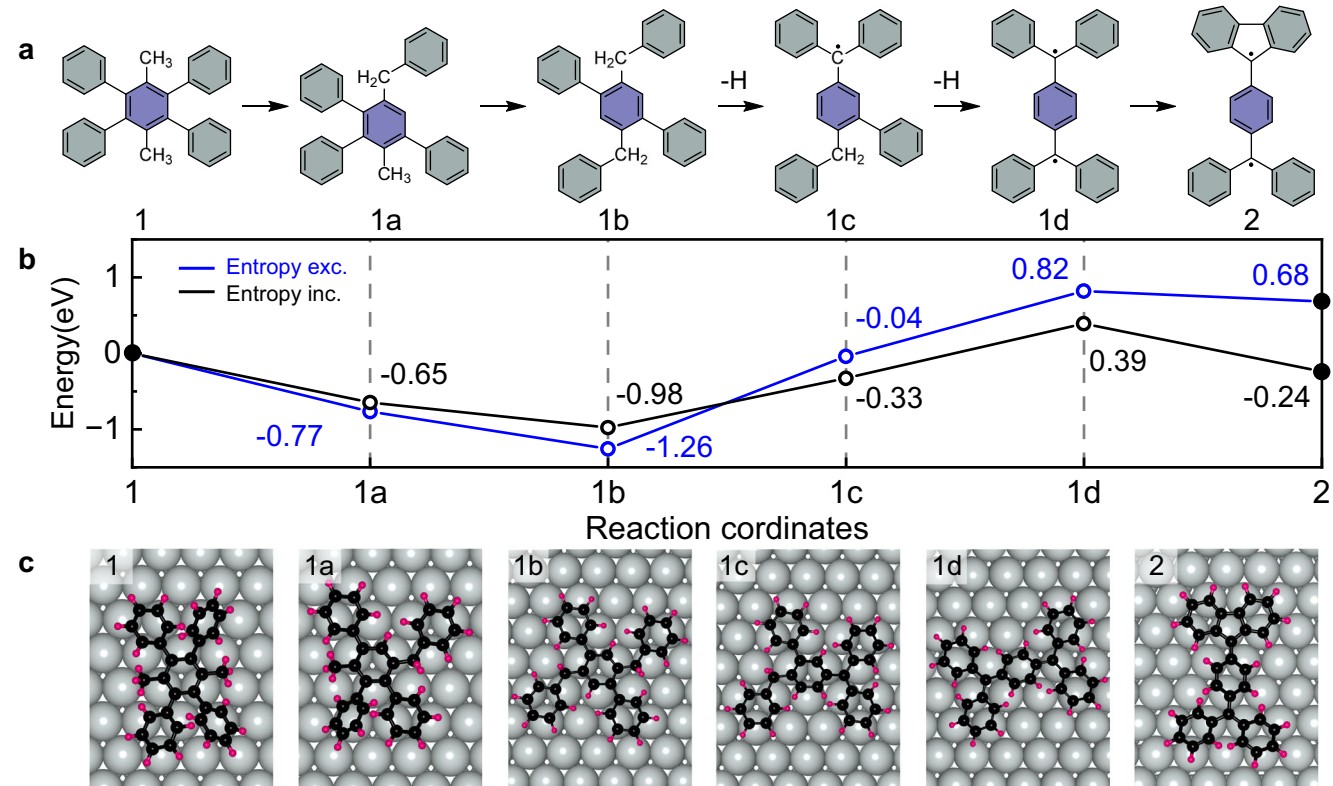

**Fig. 3 | Proposed reaction pathway of the phenyl migration reaction.**
**a** Proposed reaction pathway of the structural rearrangement of precursor **1** to form phenyl-migrated product **2** via a dehydrogenative-induced detachment and migration of phenyl groups. Shaded rings with two different colors mark the four phenyl groups and the central benzene in the precursor. **b** Energy diagram with (black) and without (blue), including enthalpic and entropic effects for the stepwise phenyl groups migrations. **c** Top views of the adsorption configurations of different intermediates shown in **a** on Au(111).

homolytic cleavage of C−H bonds in the methyl group (Supplementary Fig. 18), and then the radicals would attach the ipso position, i.e., carbon sites in the central benzene ring where phenyl groups connected, forming the intermediate (int1). Subsequently, the phenyl groups are expelled due to rearomatization. Subsequent reactions between the methyl radicals and the expelled phenyl group radicals produce a fully saturated intermediate 1a, interpreted as one phenyl group migrating to the methyl-C (Supplementary Fig. 18). The corresponding energy diagram from DMTPB precursor **1** to the phenyl migrated product **2** at 440 K is shown in Fig. 3b, which gives a

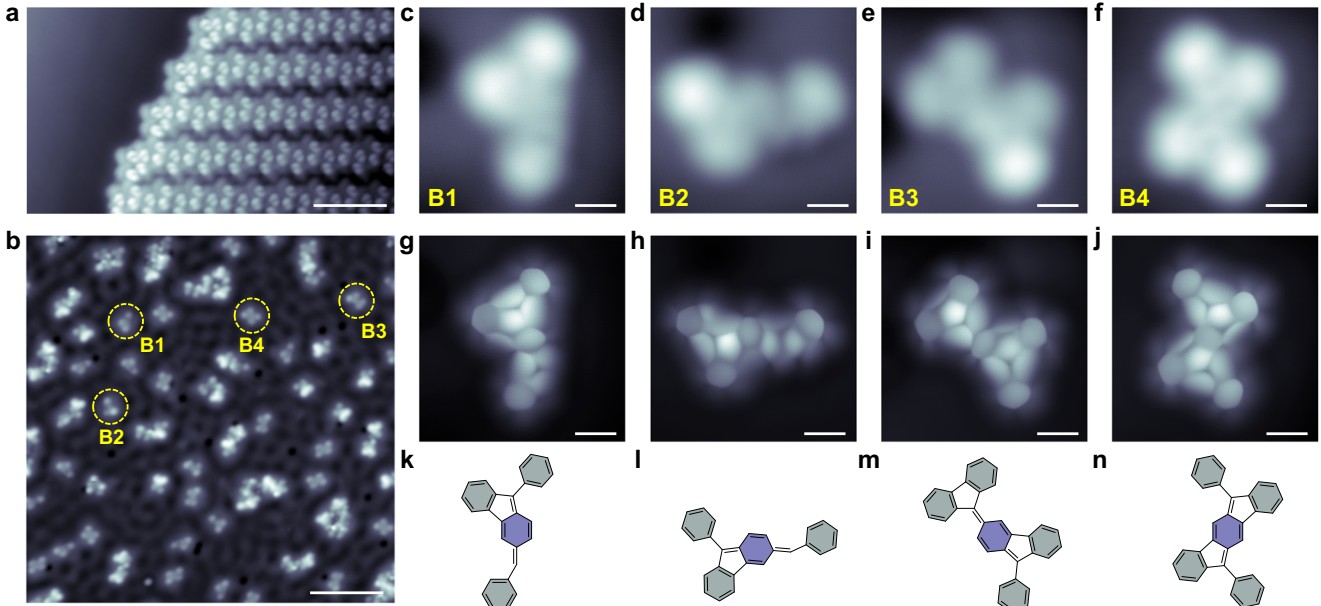

**Fig. 4 | Products of phenyl migration reaction formed on Cu(111) after annealing at 420 K. a** STM topography of DMTPB self-assembly on Cu(111). **b** Large-scale STM topography ($I$ = 50 pA, $V_s$ = 100 mV) of different products after thermal annealing of the DMTPB self-assembly on Cu(111) at 420 K. **c**–**f** STM topographies (CO tip, $I$ = 50 pA, $V_s$ = 100 mV) of major monomer products, as denoted by yellow dashed circles numbered by **B1**, **B2**, **B3** and **B4** in **a**. **g**–**j** BR-STM images (CO tip, $V_s$ = 2 mV) of products **B1**, **B2**, **B3**, **B4** correspond to **c**–**f**, respectively. **k**–**n** The identified molecular structures of five products corresponding to **B1**, **B2**, **B3**, **B4** in panels **g**–**j**. Shaded rings with two different colors mark the phenyl groups and the central benzene in the precursor. Scale bar: **a** 5 nm **b** 6 nm; **b**–**i** 0.5 nm.

reasonable interpretation of the experimental results. Importantly, the oxidative ring closure of methyl groups and the cyclodehydrogenation of adjacent phenyl groups are not energetically favored, which further supports our experimental observations. A detailed comparison of different possible competitive reactions can be found in Supplementary Fig. 19. Specifically, we plot both the energies of the intermediates with and without including enthalpic and entropic effects in Fig. 3b, together with the corresponding optimized structures of the proposed intermediates and observed products on the surface (Fig. 3c). In the transformation, the migrations of the first phenyl and the second phenyl groups result in two energetically more stable intermediates 1a and 1b, implying that the migration can be triggered upon annealing. Although the migrations of the third and the fourth phenyl groups cause an uphill energy of the intermediates, we proposed here that this two steps removal of the hydrogens at this stage will break the reversibility and thus proceed the further migration of phenyl groups. A more detailed reaction coordinate with possible intermediates is further discussed in the Supplementary Information (Supplementary Fig. 20). The migration of all phenyl groups as well as the dehydrogenation produce the intermediate 1d, which has a diradical character, referred to as Thiele's hydrocarbon, and could easily transform to the experimentally observed product **2** due to its highly reactive nature. The dehydrogenation sequence of the methyl groups is further discussed and compared in detail in Supplementary Figs. 21–25 and Supplementary Tabs. 2–5, from which the most energetically favorable path is illustrated in Fig. 3.

To further test and verify the phenyl migration reaction, we deposited the DMTPB precursor onto a Cu(111) substrate. We observed an ordered self-assembly (Supplementary Fig. 26), which is different from that on the Au(111) surface (Fig. 2a). Subsequent annealing to 420 K formed several monomer products with different yields (Supplementary Tab. 6) as well as molecular clusters, as shown in Fig. 4a. The constant-height BR-STM images (Fig. 4b–e) and corresponding identified chemical structures (Fig. 4f–i) of the monomer products denoted by yellow dashed circles (**B1** to **B4**) in Fig. 4a, which unambiguously confirmed that the phenyl group migration reaction can also

take place on the Cu(111) substrate. The chemical structures of **B1** and rare species **B5** are also complemented by nc-AFM imaging and DFT simulated results, which show good agreements (Supplementary Fig. 27). Interestingly, compared to the DMTPB precursor, the products **B1** and **B2** lost a benzene ring, revealing that the phenyl groups would be indeed expelled and the reactivity complexity of the migration of phenyl groups. Owing to the on-surface cyclodehydrogenation reaction, other products could be found after thermal treatment at increased temperature (Supplementary Fig. 28). Moreover, we also identified different products of phenyl group migration on a Ag(110) surface after annealing to 460 K (Supplementary Figs. 29 and 30). The successful observation of phenyl group migration on the Au(111), Cu(111) and Ag(110) surfaces demonstrates that the migration is a universal phenomenon and adaptable to surfaces with different catalytic activity and symmetry.

As far as we know, most radical migrations are originally discovered as side reactions. It is hard to obtain final products in a controllable manner, especially in the field of on-surface synthesis. The ultra-high yield of the intermediate product **2** (Supplementary Fig. 3j) implies that this experimentally observed phenyl group migration reaction is a precise on-surface chemical reaction with mono-selection, which can serve as a significantly efficient methodology in preparing synthetically building blocks. The migration with the absence of forming spirocyclic intermediate here also complements the present mechanism of radical aryl migration reaction. Moreover, most early reported radical migrations require the use of toxic reagents and harsh conditions, such as high temperature or high dilution, which do not meet the requirements of green chemistry[43–45], meanwhile, our results indicate that phenyl group migration reaction could take place on different catalytic surfaces under mild heat treatment without additional complexity.

In summary, we have successfully identified and visualized a phenyl group migration involving the C–C bond cleavage between phenyl groups and the precursor backbone on Au(111) substrate. The subsequent reaction produces various intermediates and products derived from the cascaded cyclodehydrogenations process on the

surface, which were directly captured by STM/nc-AFM. DFT calculations confirmed the energetically favored phenyl groups migration scenario. The universality of phenyl group migration of DMTPB precursors is also unambiguously realized on Cu(111) and Ag(110) substrates. The results here provide detailed mechanistic insights into the field of on-surface synthesis and guidance into the design and synthesis of new chemical species.

## Methods

### Sample preparation

The atomically clean Au(111), Cu(111), and Ag(110) (MaTecK) surfaces were obtained by cycles of argon-ion sputtering and annealing at 800, 785, and 770 K, respectively. The DMTPB molecules (Sigma Aldrich, 99%) were thermally deposited onto substrates by a standard Knudsen cell heating at 360 K, while the substrates were held at room temperature. CO molecules were dosed onto the sample surfaces for tip modification.

### STM and AFM measurements

Experiments were performed by using an ultra-high vacuum-low temperature-scanning tunneling microscope (UHV-LT-STM, Scienta Omicron) with a base pressure better than $1 \times 10^{-10}$ mbar. After deposition and annealing, the samples were analyzed with STM and nc-AFM at 4.2 K. All the STM images were acquired with the constant current or constant height mode by using an electrochemically etched tungsten tip, and all given voltages were applied to the sample with respect to the tip. Nanotec Electronica WSxM software[46] was used to process images shown here. nc-AFM measurements were performed at 4.2 K with tungsten tips placed on qPlus tuning fork sensors[47]. The tips were functionalized with a single CO molecule at the tip apex picked up from the CO-dosed surface. Sensor 1 was driven at its resonance frequency (28214 Hz) with a constant amplitude of 50 pm. Sensor 2 was driven at its resonance frequency (28,248 Hz) with a constant amplitude of 90 pm. The $\Delta z$ was positive (negative) when the tip-surface distance was increased (decreased) with respect to the STM set point at which the feedback loop was opened.

### Calculation details

DFT-based first-principles calculations were performed in a plane-wave formulation with the projector augmented wave method (PAW) as implemented in the Vienna Ab initio Simulation Package (VASP) code. The Perdew–Burke–Ernzerhof (PBE) parametrization of the generalized gradient approximation (GGA) was used. The cutoff energy for the plane waves was 450 eV. The vacuum layer is 20 Å between neighboring slabs. In relaxation, atoms in the bottom layer of the substrate were fixed and all the other atoms were relaxed until the atomic forces were less than 0.02 eV/Å. The Brillouin zone was sampled with only the Γ-point in all calculations. A dispersion correction of total energy (DFT-D2 method) was used to incorporate the long-range van der Waals interaction. Simulations of nc-AFM were performed with a modification of the particle probe model code by Hapala et al[48]. The structure was taken from DFT simulations, and the electrostatic potential was extracted from the Hartree potential[49]. The simulations involved oscillating a probe particle with an amplitude of 60 pm at a height of 2.88 Å above the substrate. We also used a quadrupole $dz^2$ tip model[50], and a tip lateral spring constant of about 0.25 N/m, which proved to give the best agreement with the experiment. Zero-point energy (ZPE), enthalpy, and entropy contributions to free energies at 440 K were calculated from vibrational modes of surface species, which were computed with the finite difference approach as implemented in the VaspGibbs code[51].

## Data availability

The data that supports the findings of this study are available within the paper or its Supplementary Information. The raw data are available from the corresponding author upon request.

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

## Acknowledgements
The project is financially supported by the National Natural Science Foundation of China (Nos. 62271238 and 61901200), the Yunnan Fundamental Research Projects (Nos. 202201AT070078, 202101AV070008, 202101AW070010, 202101AU070043, and 202201BE070001-009), the Strategic Priority Research Program of Chinese Academy of Sciences (XDB30000000), and the Dongguan Innovation Research Team Program. Numerical computations is performed on Hefei advanced computing center.

## Author contributions
J.C., J.L., and L.G. conceived and coordinated the research project. Z.R., S.S., and Y.Z. carried out the experiments. B.L. performed the DFT calculations under the guidance of L.G. All authors participated in discussing the data and editing the manuscript.

## Competing interests
The authors declare no competing interests.
