## [Peer Review File · Nature Communications]

Real-Space Imaging of a Phenyl Group Migration Reaction on Metal SurfacesREVIEWER COMMENTS

Reviewer #1 (Remarks to the Author):

The manuscript by Z. Ruan and coworkers reports on a very interesting case of on-surface phenyl migration, as studied by high-resolution scanning probe microscopies and DFT calculations. The precursor molecule, 1,4-dimethyl-2,3,5,6-tetraphenylbenzene, undergoes unexpected transformation into various products in which all four phenyl substituents have migrated from the central benzene ring to the methyl groups in the 1- and 4-positions. This study is a prime example for the capabilities of advanced low-temperature scanning probe methods, which use CO-functionalized tips to clarify molecular structures on the single-molecule level. There is currently no other analytic method that could clarify molecular structures in complex mixtures with similar sensitivity. One of the surprising results of this study is the high selectivity of the phenyl migration reaction. While several products are formed and their structures are unambiguously clarified, the main product is formed with 95% selectivity. To clarify the multi-step reaction mechanism, the authors provide complementary DFT calculations, from which also reaction energies and activation barriers are obtained. The phenyl migration is observed on three surfaces of different composition, structure and reactivity (Au(111), Cu(111), Ag(110)), indicating that the findings are rather universal and not specific for one surface. In the reviewer's opinion, the reported results are of outstanding relevance not only for the thriving field of on-surface synthesis, but for organic chemistry in general, because the migration of groups larger than hydrogen is not really well understood and mechanisms are often rather speculative. Compared to the standards of the field, this study is groundbreaking in its rigor. The scientific importance of the subject, combined with the very high quality of the data and the excellent preparation of the figures makes this study in principle suitable for publication in Nature Communications. The only real weakness of this manuscript is the quality of the text. Already the first paragraph of the introduction contains five spelling mistakes and it does not get much better later. In addition, grammar and wording are often awkward, making reading difficult. In the age of efficient and automated spell and grammar checkers, most of this is not excusable. Furthermore, several references in the main text to Supplementary Figures are wrong and there are Supplementary Figures that are not mentioned in the text at all, although they contain highly relevant material. All in all, the manuscript creates the impression that it lacks final polishing by the PI. In conclusion, the reviewer recommends publication of this manuscript in Nature Comm., but the above-mentioned problems must be eliminated and technical points below should be considered.

1. Based on the proposed sp³ centers in various product structures and the resulting non-planarity of the molecules, one may expect stronger contrasts in the STM and AFM images than are actually observed. Examples are 7 and 8, which look rather uniform in STM (e.g. in Suppl. Figure 4) despite the proposed sp³ centers. The reviewer believes that the authors are correct in their assignments, especially because of Suppl. Figure 5, in which experiment and simulation are compared for one example, showing excellent agreement. Nevertheless, this point should be discussed in more detail and the possibility of planar (quinoid) structures for 7 or 8 (and others) should be considered (and explicitly excluded, if applicable).
2. Lines 131-132: "observed three phenyl groups migrated products (Supplementary Fig. 5)" – Should this read "Supplementary Fig. 6"?
3. Lines 135-136: "observed an ordered self-assembly (Supplementary Fig. 7)" – Should this read "Supplementary Fig. 17"?
4. Line 144: "treatment at increased temperature (Supplementary Fig. 9)" – Should this read "Supplementary Fig. 19"?
5. Lines 145-146: "on Ag(110) at 460 K (Supplementary Fig. 13)" – Should this read

“Supplementary Fig. 21”?

Reviewer #2 (Remarks to the Author):

Cai and coworkers have presented a detailed work for how phenyl migration reaction on surfaces may be used on metal surfaces. They claim this type of reaction has not been reported on surfaces before and as far as I am concerned this is correct. Interestingly the reaction seems to work on several different surfaces. I think this type of reaction really adds as a valuable contribution to the various on-surface reaction schemes for the design of novel molecular structures. I do, however, have some questions regarding the theoretical part and reaction mechanism of the study which I think the authors need to carefully answer before the manuscript can be accepted for publication.

1. I do not quite understand Figure 3. The authors need to make it clearer for what steps they actually calculated the energy profile (and the resolution of the figures in (f) needs to be improved). To go from state 1 to 1a in Fig 3a it seems that there is a hydrogen migration and phenyl migration simultaneously. Does these processes occur in a concerted fashion with the small barrier of 0.94 eV? Or are there many individual steps and the authors show the effective barrier for each step? The same comment can be made for the other phenyl migration steps as well.

2. How can the authors be sure that both phenyls on each side of the molecule migrate? E.g. going from 1a to 1b is endothermic. And the same goes for 1c to 1d. Why would this happen? And should it not be possible to explain the reaction products on the basis that only one phenyl group migrates to each methyl group?

3. How were the activation energies calculated? Nudged elastic band method?

4. How large surface unit cell was used in the calculations? Did the authors check that the Gamma-point only k-point sampling is sufficient to provide numerically converged results?

Reviewer #3 (Remarks to the Author):

This manuscript reports on the experimental evidence of phenyl group migration in thermally-driven, on-surface reactions. Starting from a 1,4-dimethyl-2,3,5,6-tetraphenyl benzene (DMTPB) precursor deposited on Au(111), Cu(111), and Ag(110) surfaces in ultrahigh vacuum conditions, annealing of the sample induced intramolecular transformations that led to several products, structurally characterized via bond-resolved STM and noncontact AFM at low temperature. The formation of some products can only be justified assuming the migration of one or more phenyl groups from the central benzene to the methyl groups positions, followed by cyclodehydrogenation cascades. The authors report structural identification of an impressive number of observed products, with very high quality data, complemented by DFT calculations of some reaction pathways and simulation of some nc-AFM images. The reported results may certainly be of interest to the scanning probe microscopy and on-surface synthesis communities. However, the described complexity has not been fully unraveled, and I would suggest to address the following points before this manuscript can be considered for publication.

Key finding of this manuscript is the observation of reaction products ascribed to a phenyl migration process. Most of the reported products could in principle arise from a complex intramolecular reaction sequence that involves both phenyl migration and cyclodehydrogenation. For instance, considering the precursor and the products 2-6, only one, two or three phenyl groups could migrate to afford products 5, 4-and-6, and 2, respectively. This would contrast with the interpretation of four migrated phenyl groups, via the intermediate 1d, as reported by the authors. The ultimate proof of the migration of all four phenyl groups (via the intermediate 1d) is strictly interlinked with the structural validation of product 7, which is the majority product (95%) after annealing to 440 K. However, this product is imaged via constant current STM only (Figure S3b), and no unambiguous identification of its chemical structure is provided, which is instead highly desirable. Assignment of the chemical structure of 8 suggests that four phenyl groups can indeed migrate, but this may only occur in a very minority of the species. In view of the importance played by selectivity in on-surface reactions, I would suggest to carefully address this point. I am aware that the presence of sp³ carbon and twisted phenyls may render this task difficult, but constant height images (being BR-STM or nc-AFM) as function of tip-sample distance or imaging of the planar half of specie 7 could lead to the desired assignment. DFT-simulated STM or nc-AFM images could also support the experimental findings. Therefore, either the authors confirm the structure of 7 or they should modify the reaction scheme in Figure 1, where all products originate from 1d, and the related discussion throughout the text.

Figure S12: The STM image of specie 7 appears significantly different from those reported in other figures (e.g. Figure S3b and S4). The evident asymmetric lobe due to a presumably tilted phenyl group is absent, and the appearance is instead very similar to that of specie 8 reported in Figure S5a. As the scanning parameters are always the same (50 pA and 100 mV), are the authors sure about the assignment of the specie in Figure S12?

Selectivity is crucial when reporting new on-surface reactions. The information that after annealing to 440 K 95% of the products are in form of specie 7 is pivotal, and I would suggest moving it to the main text (upon unambiguous assignment of the chemical structure of this specie). This discussion could then be followed by the observations of all other products after annealing to 590 K.

The energy diagram in Figure 3 and the related discussion requires some attention. Three competing processes could in principle be involved in the intramolecular transformations of the precursor: (i) cyclodehydrogenation of two adjacent phenyl groups, (ii) C-C bond formation between a phenyl and a methyl group, (iii) phenyl group detachment and re-attachment onto a methyl group (referred to as phenyl migration). In the absence of a proper validation of structure 7 (see previous comments), the observed products 2-6 can be regarded as arising from a combination of processes (ii) and (iii). However, it is clear that process (i) does not occur. Now, a comparison of the calculated energy diagrams of the two processes (ii) and (iii) would be beneficial to unravel subtle mechanistic details. In particular, it would be instructive to compare the reaction sequence 1>TS1>1a (representative of process (iii)) with a reaction sequence representative of process (ii), where a phenyl group binds an activated methyl. Moreover, the sequence 1>TS1>1a reported in Figure 3 entails a C-C bond scission upon phenyl detachment, which should be energetically demanding, and not included in the diagram. Could the authors perform nudged elastic band (NEB) calculations of this reaction step to have more insights into the phenyl migration vs phenyl attack of activated methyl?

How can the obtainment of specie 3 not arise from direct C-C coupling from 1? The activated

methyl group (CH₂*) could directly be attacked by a neighboring phenyl and ultimately lead to specie 3 without implying any methyl migration. Can the author elaborate on that and provide further sup-port for one or the other pathway?

Why is the Au(111) surface held at 320 K during deposition of the precursors? Was room temperature somehow not suited?

Some of the observed reaction products are assigned to chemical structures comprising sp³ carbon atoms with H atoms pointing away from the substrate (e.g. species 2, 8, C1-C4, and others). While this feature is usually well-visible by height-dependent nc-AFM imaging, BR-STM may fail in capturing such subtle detail. Figure S5 reports nc-AFM investigation of specie 8. However, no clear fingerprint of such feature is observed. Could the authors comment on this?

Figure S5: The model should comprise two H atoms pointing away from the surface, located at the five-membered rings apexes close to the central phenylene. They seem to be absent in the top view image of the optimized chemical model.

Figure S13: There seems to be confusion with the 1' and 1'' labeling. Does 1A coincide with TS1?

Page 7, line 129: Product 1c is not the most energetically favored, according to Figure 3f, but rather 1a.

Some of the discussions reported in the manuscript are hard to read, and a careful revision of the English would be desirable. Moreover, there are some mistakes in the numbering of figures (especially some supplementary figures referred to at page 7) and some typos through the text (e.g. "Methords") that should be fixed.

Point-by-point Response to NCOMMS-21-44910

We thank the reviewers for their constructive comments and suggestions. We have performed extensive additional work, especially structural elucidations and DFT calculations. We have addressed all the comments point-by-point and revised the manuscript accordingly. In this response letter, comments from the reviewers are summarized in blue typeface and our detailed responses are in regular black typeface. Our changes to the text are in *italic* typeface.

=====

Reviewer 1:

The manuscript by Z. Ruan and coworkers reports on a very interesting case of on-surface phenyl migration, as studied by high-resolution scanning probe microscopies and DFT calculations. The precursor molecule, 1,4-dimethyl-2,3,5,6-tetraphenylbenzene, undergoes unexpected transformation into various products in which all four phenyl substituents have migrated from the central benzene ring to the methyl groups in the 1- and 4-positions. This study is a prime example for the capabilities of advanced low-temperature scanning probe methods, which use CO-functionalized tips to clarify molecular structures on the single-molecule level. There is currently no other analytic method that could clarify molecular structures in complex mixtures with similar sensitivity. One of the surprising results of this study is the high selectivity of the phenyl migration reaction. While several products are formed and their structures are unambiguously clarified, the main product is formed with 95% selectivity. To clarify the multi-step reaction mechanism, the authors provide complementary DFT calculations, from which also reaction energies and activation barriers are obtained. The phenyl migration is observed on three surfaces of different composition, structure and reactivity (Au(111), Cu(111), Ag(110)), indicating that the findings are rather universal and not specific for one surface. In the reviewer's opinion, the reported results are of outstanding relevance not only for the thriving field of on-surface synthesis, but for organic chemistry in general, because the migration of groups larger than hydrogen is not really well understood and mechanisms are often rather speculative. Compared to the standards of the field, this study is groundbreaking

in its rigor. The scientific importance of the subject, combined with the very high quality of the data and the excellent preparation of the figures makes this study in principle suitable for publication in Nature Communications.

Author reply: We thank the reviewer very much for the time devoted to our manuscript, the very high appreciation of our work recommending publication, and his/her insightful comments, which have been replied point-to-point below, altogether reinforcing the quality of our manuscript.

Comment: 1) The only real weakness of this manuscript is the quality of the text. Already the first paragraph of the introduction contains five spelling mistakes and it does not get much better later. In addition, grammar and wording are often awkward, making reading difficult. In the age of efficient and automated spell and grammar checkers, most of this is not excusable. Furthermore, several references in the main text to Supplementary Figures are wrong and there are Supplementary Figures that are not mentioned in the text at all, although they contain highly relevant material. All in all, the manuscript creates the impression that it lacks final polishing by the PI. In conclusion, the reviewer recommends publication of this manuscript in Nature Comm., but the abovementioned problems must be eliminated and technical points below should be considered.

Author reply: We apologize for the poor language of our manuscript. We have carefully revised and polished the whole manuscript and supplementary materials. We also update the referred supplementary figures in the revised manuscript to make sure that it can give access to sufficient details of the relevant text. Now we believe the revised paper could provide a more readable description of the main results of this study.

Comment: 2) Based on the proposed sp^3 centers in various product structures and the resulting non-planarity of the molecules, one may expect stronger contrasts in the STM and AFM images than are actually observed. Examples are 7 and 8, which look rather uniform in STM (e.g. in Suppl. Figure 4) despite the proposed sp^3 centers. The reviewer believes that the authors are correct in their assignments, especially because of Suppl. Figure 5, in which experiment and simulation are compared for one example, showing excellent agreement. Nevertheless, this

point should be discussed in more detail and the possibility of planar (quinoid) structures for 7 or 8 (and others) should be considered (and explicitly excluded, if applicable).

Author reply: We appreciate the reviewer's constructive suggestions. Indeed, the hydrogen atoms pointing out the substrate surface in the sp^3 carbon would result in stronger contrasts in the STM and AFM images. The absence of such contrast in the STM images of these cascaded cyclodehydrogenation intermediates makes us reconsider our assignments. Since most of the intermediates are generated by product **8** via simple cyclodehydrogenation reactions, we first review the chemical structures of product **8**. Our comprehensive nc-AFM simulations of three possible chemical structures, i.e., hydrogen atoms saturated form, canonical diradical, or quinoid structures, indicate that product **8** is in the form of diradical hydrocarbon, as can be judged from a comparison of the experiment results and simulations in Figure R1 as follows.

- (i). In the experimental nc-AFM images obtained at a moderate tip-sample separation, the outer edges (denoted by yellow arrows in Figure R1f) of the fluorenes are well resolved, which coincides with the simulated nc-AFM image (Figure R1f), while such features are absent in the saturated form in Figure R1c. At a relatively close tip-sample separation, although the edges are also identified for the hydrogen saturated form, the fluorene groups are distorted, which is not in accordance with the experimental results.
- (ii). The apex of the five-membered rings (denoted by the orange arrow in Figure R1f) are well resolved in the experimental nc-AFM image, which agrees with the simulations. However, it shows a darker feature for the hydrogen saturated form at a moderate tip-sample separation (Figure R1c). These dark features originate from the hydrogen atom at the sp^3 center, and are identified as bright protrusions at larger tip-sample separations.
- (iii). The quinoid structure can be directly ruled out due to its complementary planar character, as can be seen in Figure R1g-1i.

Figure R1. Three possible chemical structures (a, d, g), optimized adsorption configurations (b, e, h), nc-AFM images (f), and simulated nc-AFM images (c, f, i) of product **8**.

After the identification of product **8**, we now back to product **7**, which is the first product observed after increasing the annealing temperature. A comprehensive comparison of the experimental nc-AFM images and simulated nc-AFM images indicates that product **7** is also in the diradical form, which does not contain the hydrogen atoms at the sp^3 center of the apex of the five-membered ring. In the simulated nc-AFM images of the diradical form for product **7**, the twisted phenyl rings are well reproduced, as indicated by orange, green and red arrows in Figures R2f, R2k and R2l. while such features are absent in the H saturated form. Moreover, the hydrogen atoms in the sp^3 center give pronounced features (Figure R2o-q, white arrow) which are absent in the experimental nc-AFM images (Figure R2c-f). Although the features of the twisted phenyl groups are also visible and hydrogens pointing out the surface disappear at a very close tip-sample separation (Figure R2r), this would lead to a distortion of the molecule and a considerable background signal, which is strikingly different from our experiments. We also rule out the possibility of the quinoid form of product **7**, since this would lead to a planar configuration (Figure R1g-i) and contradict the observation in Figure R2c-f, where the nonplanar feature is identified. In conclusion, we assign product **7** to the diradical form, as shown in Figure R2g.

Figure R2. Structure elucidation of product **7**. (a) Experimental STM, (b) BR-STM and (c-f) nc-AFM images of product **7**. (g, m) Two possible chemical structures. (h, n) Optimized adsorption configurations and (i-l, o-r) a series of simulated nc-AFM images of product **7**.

We also obtained the ultra-high-resolution BR-STM images of product **14** which does not contain the non-planar phenyl groups, as shown in Figure R3. The absence of any feature at the sites denoted by orange arrows in Figures R3b and 3c indicates hydrogen atoms are not involved. We thus rule out the hydrogen saturated form. The possible canonical structure of product **14** by removing the two hydrogen atoms are listed in Figure R3d. The absence of hydrogen atoms is also valid for the other intermediates, as can be seen in the series of high-resolution BR-STM/nc-AFM images shown in the supplementary materials.

Figure R3. Structure confirmation for product **14**. (a) Constant current STM image and (b) High-resolution BR-STM image and (c) Zoom-in BR-STM image of the area in (b) marked by an orange square. (d) Possible structures of product **14**.

At this point, it's noteworthy to furtherly discuss the resonance structure of the cascaded cyclodehydrogenation intermediates. The cyclodehydrogenation intermediates **2**, **9**, and **14** can be simply regarded as hydrocarbons with a *p*-quinodimethane core (*p*-QDM) [Acc. Chem. Res. 2017, **50**, 1997-2006; Physical Organic Chemistry of Quinodimethanes, 2018, 107-168. J. Am. Chem. Soc. 2013, **135**, 6363-6371; Nat. Commun. 2021, **12**, 1-7], as shown in Figure R4. Considering that product **2** is largely present at 590 K (27%) as well as it can be found at a higher temperature, which is not in line with its highly reactive diradical character. We thus propose that product **2** may have a closed-shell ground state with four sextets at the corners. This can be also evidenced by a shorter bond length between the fluorene and central benzene ring, as denoted by the yellow arrow in Figure R4c. Similar short bond length is also observed for product **9** and product **14**, as shown in Figures R4d and R4e. A dominant closed-shell ground state thus is also expected here.

Figure R4. Chemical structures of products **2**, **9** and **14**. (a) Schematic illustration of possible canonical forms of considered structures. (b, d) STM and (c, e) nc-AFM images of product **2** and **9**, respectively. (f) STM and (g) BR-STM images of product **14**.

As for cascaded cyclodehydrogenation intermediates **12/C4**, **13**, **15**, **16** and **16'**, although the additional formed five-membered ring may alter the conjugation, for the sake of simplicity, we here still regard them as indeno[1,2-b]fluorene derivatives. According to literature reports [J. Phys. Chem. A 2015, **119**, 10620-10627; J. Am. Chem. Soc. 2019, **141**, 12346-12354; Nat. Chem. 2016, **8**, 753-759], indeno[1,2-b]fluorene has a canonical structure with diradical character $y = 0.072$, which means it has a dominant contribution of the closed-shell structure and indeno[2,1-b]fluorene has more open-shell character with $y = 0.645$. Note that product **15** can be also drawn as indeno[2,1-c]fluorene derivative ($y = 0.021$), as shown in Figure R5c, which also has a dominant closed-shell character. Based on the above discussion, we give the dominant resonance structure of these products in Supplementary Fig. 11, and insert the corresponding discussion of the resonance form into the supplementary materials.

Figure R5. Resonance structures of products **12/C4**, **13**, **15**, **16** and **16[•]**. (a-c) different indenofluorene regioisomers (top) and the corresponding classified products based on the indenofluorene core proposed (bottom).

Additionally, we assign the fully reacted products of nanographene **3-6** to the closed-shell canonical structure with four sextets as the BR-STM images and their corresponding STS do not show any feature near the Fermi level, while the open-shell diradical form often gives rise to detectable features [J. Am. Chem. Soc. 2020, **142**, 1147-1152; Sci. Adv. 2019, **5**, eaav7717. Angew. Chem. Int. Ed. 2021, **60**, 16208-16214]. Besides the STM-based measurements, in the nc-AFM images, the shorter bond length (yellow arrows) and brighter contrast (blue arrows) in Figure R6 indicate a higher bond order of these products, which may also explain their double bond character.

Figure R6. Chemical structures of products **3**, **4**, **5**, **6**. (a) nc-AFM images of products 3-6 and (b) their chemical structures, respectively.

The hydrogen saturation of products on the Cu(111) substrate is also confirmed. Our simulated nc-AFM images show the excellent agreement with our experimental results, as shown in Figure R7. We thus propose that the products on Cu(111) have quinoid configurations. Other products on the Cu(111) and Ag(110) substrate as shown in the supplementary materials have been modified based on the above discussions.

Figure R7. Chemical structure elucidation of product **C2** on Cu(111). (a) STM image, (b) BR-STM image, (c) optimized adsorption configuration and (d) proposed chemical structure for product **C2**, respectively. (e) Experimental and (f-h) simulated nc-AFM images of product **C2** with decreased tip height.

In summary, we propose that all the experimentally observed products on Au(111), Cu(111) and Ag(110) do not contain the hydrogen atoms at sp^3 sites. A dominant diradical form or canonical quinoid form is responsible for the observed products. All the mentioned products on the three substrates, i.e., Au(111), Cu(111) and Ag(110), have been now updated according to the above discussion, as shown below in Table R1. The above discussions have been inserted into the Supplementary information as Supplementary Figs. 8-9 and Figs. 14-17 along with the related description.

Table R1. Summarized chemical structures of observed products on the Au(111), Cu(111) and Ag(110) substrates

Au(111)						
	Product 2	Product 3	Product 4	Product 5	Product 6	Product 7
	Product 8	Product 9	Product 10	Product 11	Product 12	Product 13
	Product 14	Product 15	Product 16	Product 16	Product 17	Product 18
Product 19						
Cu(111)						
	Product C1	Product C2	Product C3	Product C4	Product C5	Product C6
Product C7	Product C8	Product C9	Product C10	Product C11		

Ag(110)						
	Product A1	Product A2	Product A3	Product A4	Product A5	

Comment: 3) Lines 131-132: “observed three phenyl groups migrated products (Supplementary Fig. 5)” – Should this read “Supplementary Fig. 6”?

Author reply: We apologize for this mistake. We have revised the manuscript and the supplementary materials and corrected this incorrect reference and also the migrated number “three”. The sentence “observed three phenyl groups migrated products (Supplementary Fig. 5)” now has been delete in our revised manuscript, and now this product is inserted into Supplementary Fig. 11 as product **17**.

Comment: 4) Lines 135-136: “observed an ordered self-assembly (Supplementary Fig. 7)” – Should this read “Supplementary Fig. 17”?

Author reply: We apologize for this mistake. Since we inserted additional Supplementary Figures into the supplementary materials, we change this sentence to “observed an ordered self-assembly (Supplementary Fig. 28)”.

Comment: 5) Line 144: “treatment at increased temperature (Supplementary Fig. 9)” – Should this read “Supplementary Fig. 19”?

Author reply: We apologize for this mistake. Since we inserted additional Supplementary Figures into the supplementary materials, we changes this sentence to “treatment at increased temperature (Supplementary Fig. 30)”.

Comment: 6) Lines 145-146: “on Ag(110) at 460 K (Supplementary Fig. 13)” – Should this read “Supplementary Fig. 21”?

Author reply: We apologize again for this mistake. The referred figure now has been corrected to Supplementary Fig. 32 in our revised manuscript along with the supplementary materials.

Reviewer 2:

Cai and coworkers have presented a detailed work for how phenyl migration reaction on surfaces may be used on metal surfaces. They claim this type of reaction has not been reported on surfaces before and as far as I am concerned this is correct. Interestingly the reaction seems to work on several different surfaces. I think this type of reaction really adds as a valuable contribution to the various on-surface reaction schemes for the design of novel molecular structures. I do, however, have some questions regarding the theoretical part and reaction mechanism of the study which I think the authors need to carefully answer before the manuscript can be accepted for publication.

Author reply: We thank the reviewer's positive comments to our work.

Comment: 1) I do not quite understand Figure 3. The authors need to make it clearer for what steps they actually calculated the energy profile (and the resolution of the figures in (f) needs to be improved). To go from state 1 to 1a in Fig 3a it seems that there is a hydrogen migration and phenyl migration simultaneously. Does these processes occur in a concerted fashion with the small barrier of 0.94 eV? Or are there many individual steps and the authors show the effective barrier for each step? The same comment can be made for the other phenyl migration steps as well.

Author reply: We appreciate the reviewer for this important remark. We are very sorry that Figure 3 is not clear enough. We carefully analysed the reaction steps and made a new Figure 3 (Figure R8) as shown below. In the new Figure 3 (Figure R8), the 1 and 1a used before is renamed as I and II respectively. To go from state I to II, there are two individual steps: the hydrogen migration (I-Ia) and the phenyl migration (Ia-II), which barriers are 1.63 and 2.10 eV respectively. For II-III, III-IV and IV-V, there are also two individual steps.

Figure R8. (a) Proposed reaction pathway of the structural rearrangement of precursor I to form intermediate V. (b) Energy diagram of the proposed step-by-step phenyl group migration reaction pathways. (c) Top views of the adsorption configurations of different intermediates shown in (a) on Au(111).

Comment: 2) How can the authors be sure that both phenyls on each side of the molecule migrate? E.g. going from 1a to 1b is endothermic. And the same goes for 1c to 1d. Why would this happen? And should it not be possible to explain the reaction products on the basis that only one phenyl group migrates to each methyl group?

Author reply: We thank the reviewer for his/her constructive comments. Our sequential annealing steps show that all of the four phenyl groups migrate to the methyl groups in our firstly observed migration product **7**, as displayed below in Figure R9. The identified chemical structure and its very high yield (>95%) indicate that all the four phenyl groups would indeed migrate to the methyl groups in the reaction. The reaction products at increased annealing temperatures thus originate from the four-phenyl-migrated intermediates.

Figure R9. Main product after thermal treatment of the DMTPB on Au(111) substrate. (a-c) STM images of the observed product **7** with ultra-high yield (>95%). (d) Schematic illustration of the migration of phenyl groups to product **7**. (e) Constant current STM image of product **7**.

As for 1a-1b (renamed as II-III in the new Figure 3) and 1c-1d (renamed as IV-V in the new Figure 3) are endothermic, we thank the reviewer for bringing up this important point. We further investigated the sequence of the four phenyl group migrations. The optimal sequence of the four phenyl group migration is marked in red in Figure R10.

Figure R10. Calculated the optimal sequence of the four phenyl group migrations that lead from I to V. The structure of each step is represented within the rectangle, and the red phenyl and black arrows indicate the phenyl groups and directions in which the structure will migrate. The configuration of the lowest energy of products is marked in red.

However, III-IV and IV-V are still endothermic. Meanwhile, we found that there is no hydrogen on the methyl group of the reaction product V, as can be concluded from the comparison of the experimental mc-AFM images and the simulated nc-AFM images of product **7** in different forms (Figure R11). Considering that the extra hydrogen atoms are going to get away, endothermic reactions are possible to happen.

Figure R11. Structure elucidation of product **7**. (a) Experimental STM, (b) BR-STM and (c-f) nc-AFM images of product **7**. (g, m) Two possible chemical structures. (h, n) Optimized adsorption configurations and (i-l, o-r) a series of simulated nc-AFM images for product **7**.

As shown in Figure R12-15, four possibilities were considered: path 1 that dehydrogenations precede phenyl groups migrations (Figure R12); path 2 that dehydrogenations occur during I-II and II-III (Figure R13); path 3 that dehydrogenations occur during III-IV and IV-V (Figure R14); path 4 that dehydrogenations occur after two phenyl groups migrations (Figure R15). Energy diagram comparing all four plausible reaction mechanisms that lead from I to V are shown in Figure 16, which reveals that path 3 (dehydrogenating during III-IV and IV-V) is the energetic favourable reaction path. Based on the above calculated results, we made a new Figure 3 (Figure R8) as shown above. I-II and II-III are exothermic, while III-IV and IV-V are endothermic. Since the extra hydrogen atoms are going to get away during III-IV and IV-V, the reverse reaction conditions are broken. Therefore, endothermic reactions III-IV and IV-V are possible to happen. We have inserted the Figures R12-16 into the Supplementary information as Supplementary Figures 23-27 along with the related description into the revised manuscript.

Figure R12. (a) Calculated energy diagram for the stepwise phenyl groups migrations in path 1: dehydrogenations precede phenyl groups migrations. (b) Schematic illustration of the stepwise phenyl groups migrations from I to V and (c) Optimized configurations in (a).

Figure R13. (a) Calculated energy diagram for the stepwise phenyl groups migrations in path 2: dehydrogenations occur during I-II and II-III. (b) Schematic illustration of the stepwise phenyl groups migrations from I to V and (c) Optimized configurations in (a).

Figure R14. (a) Calculated energy diagram for the stepwise phenyl groups migrations in path 3: dehydrogenations occur during III-IV and IV-V. (b) Schematic illustration of the stepwise phenyl groups migrations from I to V and (c) Optimized configurations in (a).

Figure R15. (a) Calculated energy diagram for the stepwise phenyl groups migrations in path 4: dehydrogenations occur after two phenyl groups migrations. (b) Schematic illustration of the stepwise phenyl groups migrations from I to V and (c) Optimized configurations in (a).

Figure R16. Energy diagram comparing all four plausible reaction mechanisms that lead from I to V. Path 1: dehydrogenations precede phenyl groups migrations. Path 2: dehydrogenations occur during I-II and II-III. Path 3: dehydrogenations occur during III-IV and IV-V. Path 4: dehydrogenations occur after two phenyl groups migrations.

Comment: 3) How were the activation energies calculated? Nudged elastic band method?

Author reply: The activation energies in Fig. 3 were calculated via combining the nudged elastic band (NEB) method with the constrained geometry optimization method, which is used for long series of reaction steps [Nat. Synth. 2022, **1**, 289-296]. In the reaction steps I-Ia and Ia-II, we resorted to NEB calculations for each reaction steps. The calculated results for I-Ia and Ia-II are summarized in Figures R17 and R18, in which the energy barriers are 1.63 eV and 2.10 eV, respectively. The atomic configurations corresponding to the highest energy points for I-Ia and Ia-II are shown as TS1 and TS2 respectively. Due to the large number of atoms involved, relying on the nudged elastic band (NEB) for such a long series of reaction steps would have a computational cost that we cannot afford. Therefore, for the reaction steps II-IIIa, IIIa-III, III-IVa, IVa-IV, IV-Va and Va-V, we used the constrained geometry optimization method for each step to obtain a reasonable estimate of the reaction barriers. The intermediate geometries are summarised in Figure R19. Each intermediate geometry is optimized without constraints. The energy profile resulting from such a series of constrained geometry optimizations, where all atomic degrees of freedom are allowed to relax subject to the constraint of the collective variable, provides a reasonable estimate of the reaction barrier. We have inserted the Figures R17-19 into the Supplementary information as Supplementary

Figures 19, 21, 22 along with the related description into the revised manuscript.

Figure R17. Energy barriers for I to Ia from the NEB calculation.

Figure R18. Energy barriers for Ia to II from the NEB calculation.

Figure R19. Optimized configurations of the transition states TS1-TS8 which is corresponding to the highest energy points for I-Ia, Ia-II, II-IIIa, IIIa-III, III-IVa, IVa-IV, IV-Va and Va-V respectively in the energy diagram shown in Figure 3.

Comment: 4) How large surface unit cell was used in the calculations? Did the authors check that the Gamma-point only k-point sampling is sufficient to provide numerically converged results?

Author reply: A surface unit cell of $25 \text{ \AA} \times 20 \text{ \AA} \times 30.0 \text{ \AA}$ was used in the calculations. We followed the reviewer's suggestions and did more calculations by using $3 \times 3 \times 1$ k-point sampling for comparison to evaluate the feasibility of the Gamma-point only k-point sampling we used. Taking the DMTPB on Au(111) surface as an example, the optimized configurations by using $1 \times 1 \times 1$ and $3 \times 3 \times 1$ k-point samplings are shown in the Figure R20. The total energies of the DMTPB on Au(111) surface by using $1 \times 1 \times 1$ and $3 \times 3 \times 1$ k-point samplings are -4.581 eV/atom and -4.585 eV/atom respectively. Since the adsorption configurations and the total energies of the DMTPB on Au(111) surface obtained by using $1 \times 1 \times 1$ and $3 \times 3 \times 1$ k-point samplings are almost the same, we think results obtained by using Gamma-point only k-point

sampling are reliable.

Figure R20. Optimised configurations of the DMTPB on Au(111) surface by using $1 \times 1 \times 1$ (a) and (b) $3 \times 3 \times 1$ k-point samplings.

Reviewer 3:

This manuscript reports on the experimental evidence of phenyl group migration in thermally-driven, on-surface reactions. Starting from a 1,4-dimethyl-2,3,5,6-tetraphenyl benzene (DMTPB) precursor deposited on Au(111), Cu(111), and Ag(110) surfaces in ultrahigh vacuum conditions, annealing of the sample induced intramolecular transformations that led to several products, structurally characterized via bond-resolved STM and noncontact AFM at low temperature. The formation of some products can only be justified assuming the migration of one or more phenyl groups from the central benzene to the methyl groups positions, followed by cyclodehydrogenation cascades. The authors report structural identification of an impressive number of observed products, with very high quality data, complemented by DFT calculations of some reaction pathways and simulation of some nc-AFM images. The reported results may certainly be of interest to the scanning probe microscopy and on-surface synthesis communities. However, the described complexity has not been fully unraveled, and I would suggest to address the following points before this manuscript can be considered for publication.

Author reply: We thank the reviewer's positive comments to our work. We have addressed point-to-point his/her comments below, which have contributed to the improvement of our manuscript.

Comment: 1) Key finding of this manuscript is the observation of reaction products ascribed to a phenyl migration process. Most of the reported products could in principle arise from a complex intramolecular reaction sequence that involves both phenyl migration and cyclodehydrogenation. For instance, considering the precursor and the products 2-6, only one, two or three phenyl groups could migrate to afford products 5, 4-and-6, and 2, respectively. This would contrast with the interpretation of four migrated phenyl groups, via the intermediate 1d, as reported by the authors. The ultimate proof of the migration of all four phenyl groups (via the intermediate 1d) is strictly interlinked with the structural validation of product 7, which is the majority product (95%) after annealing to 440 K. However, this product is imaged via constant current STM only (Figure S3b), and no unambiguous identification of its chemical structure is provided, which is instead highly desirable. Assignment of the chemical structure

of **8** suggests that four phenyl groups can indeed migrate, but this may only occur in a very minority of the species. In view of the importance played by selectivity in on-surface reactions, I would suggest to carefully address this point. I am aware that the presence of sp³ carbon and twisted phenyls may render this task difficult, but constant height images (being BR-STM or nc-AFM) as function of tip sample distance or imaging of the planar half of specie **7** could lead to the desired assignment. DFT simulated STM or nc-AFM images could also support the experimental findings. Therefore, either the authors confirm the structure of **7** or they should modify the reaction scheme in Figure 1, where all products originate from **1d**, and the related discussion throughout the text.

Author reply: We thank the reviewer for the careful inspection of our data and his/her constructive suggestions. We have performed constant-height BR-STM and nc-AFM measurements for product **7**, as shown below in Figure R21. From the BR-STM image of the planar part of product **7** in Figures R21c and 21f, together with the nc-AFM image in Figure R21h, i, g, we confirm the planar fluorene group configuration in product **7**. According to the assignment of product **8** in Figure R1, product **7** would also have the canonical structure by removing two hydrogen atoms at the carbon atoms in methyl groups, as shown in Figure R21j. Alternatively, the two hydrogen atoms may be eliminated in the transformation of the precursor molecule. To amplify this, we obtain three nc-AFM images as a function of the tip-sample separation, together with the simulated nc-AFM images of dehydrogenated and hydrogenated structures for comparison, as shown in Figure R22.

In the constant current STM image (Figure R22a), height-dependent nc-AFM images (Figure R22c-e) and a simultaneously acquired current image (Figure R22b), the bright contrast ascribed to the top-right twisted phenyl group is observed. This feature is well reproduced in the simulated nc-AFM images of the diradical form (Figure R22i-l), as denoted in Figure R22k at a moderate tip-sample separation. The features caused by the top-left and central benzene marked by green and red arrows in Figure R22f are also reproduced in the simulated nc-AFM images (Figure R22k and 22l). We also notice that in our experiment the twist on the top-left and central benzene rings is much smaller than in this DFT-relaxed structure. This twist is affected by the registry of the molecule with the substrate, which might be different in the

experimental and simulated images. The twist of the central benzene ring can be also judged from nc-AFM images in Figures R21h and R21i, where bright protrusions are clearly observed. As for the simulated nc-AFM images (Figure R22o-r) for the sp³ structure with two hydrogen atoms, the twisted phenyl groups on the top are not captured at a tip-sample separation comparable to experiments (only the outmost feature is identified).

Figure R21. Structure elucidation of product **7**. (a) Constant current STM image, (b, c) Zoom-in STM and BR-STM image of the planar part of product **7**, respectively. (d) STM, (e) zoom-in STM (f) BR-STM and (g) nc-AFM images of the planar part of product **7**. (h, i) nc-AFM images of the lower part of product **7**, confirming the nonplanar central. (j) The corresponding chemical structure of **7**. It should be noted that (a-c) and (d-i) are taken from two different product **7**. A gentle distortion of nc-AFM image (i) was caused by drift, which does not affect the feature arising from the molecular adsorption height.

Moreover, the hydrogen atoms give pronounced features which are absent in the experimental nc-AFM images (Figure R21g-i and Figure R22c-f). Although the features of the twisted phenyl groups are visible and hydrogens pointing out the surface disappear at a very close tip-sample separation, this would lead to a distortion of the molecule and a considerable

background signal, which is strikingly different from our experiments. Here, we also rule out the possibility of the quinoid form of product **7**, since this would lead to a planar configuration (Figure R1g-i) and contradict the observation in Figure R21g and R21h as well as Figure R22f (red arrow), where the nonplanar feature is identified. In conclusion, we assign product **7** to the diradical form, as shown in Figure R22g.

Figure R22. Structure elucidation of product **7**. (a) Experimental STM, (b) BR-STM and (c-f) nc-AFM images of product **7**. (g, m) two possible chemical structures, (h, n) optimized adsorption configurations and (i-l, o-r) a series of simulated nc-AFM images of product **7**.

Comment: 2) Figure S12: The STM image of specie **7** appears significantly different from those reported in other figures (e.g. Figure S3b and S4). The evident asymmetric lobe due to a presumably tilted phenyl group is absent, and the appearance is instead very similar to that of specie **8** reported in Figure S5a. As the scanning parameter are always the same (50 pA and 100 mV), are the authors sure about the assignment of the specie in Figure S12?

Author reply: We apologize for this confusion caused by our mistake. The specie **7** shown in Figure S12 is actually specie **8**. We have addressed this mistake in our revised Supplementary Materials.

Comment: 3) Selectivity is crucial when reporting new on-surface reactions. The information that after annealing to 440 K 95% of the products are in form of specie 7 is pivotal, and I would suggest moving it to the main text (upon unambiguous assignment of the chemical structure of this specie). This discussion could then be followed by the observations of all other products after annealing to 590 K.

Author reply: We agree with the reviewer that the observed very high yield of specie 7 at 440 K is a crucial point. We have inserted the below discussion of the selectivity of specie 7 at 440 K in the revised manuscript.

(lines 76-78)“*We then slowly increased the temperature to 440 K and annealed the sample to trigger the intramolecular reaction. The ordered self-assembly transformed into individual nonplanar monomers with a very high yield (>95 %) of the major product (Supplementary Fig.3).*”

Comment: 4) The energy diagram in Figure 3 and the related discussion requires some attention. Three competing processes could in principle be involved in the intramolecular transformations of the precursor: (i) cyclodehydrogenation of two adjacent phenyl groups, (ii) C-C bond formation between a phenyl and a methyl group, (iii) phenyl group detachment and re-attachment onto a methyl group (referred to as phenyl migration). In the absence of a proper validation of structure 7 (see previous comments), the observed products 2-6 can be regarded as arising from a combination of processes (ii) and (iii). However, it is clear that process (i) does not occur. Now, a comparison of the calculated energy diagrams of the two processes (ii) and (iii) would be beneficial to unravel subtle mechanistic details. In particular, it would be instructive to compare the reaction sequence $1 > TS1 > 1a$ (representative of process (iii)) with a reaction sequence representative of process (ii), where a phenyl group binds an activated methyl. Moreover, the sequence $1 > TS1 > 1a$ reported in Figure 3 entails a C-C bond scission upon phenyl detachment, which should be energetically demanding, and not included in the diagram. Could the authors perform nudged elastic band (NEB) calculations of this reaction step to have more insights into the phenyl migration vs phenyl attack of activated methyl?

Author reply: We thank the reviewer for pointing out this aspect. We followed the reviewer's

suggestions and did more calculations for the three competing processes: (i) cyclodehydrogenation of two adjacent phenyl groups, (ii) C-C bond formation between a phenyl and a methyl group, (iii) phenyl group migration. As shown in Figure R23, the process (iii) is energetically favorable related to the process (i) and (ii). We have inserted the Figure R23 into the Supplementary information as Supplementary Figure 20 along with the related description into the revised manuscript.

Figure R23. (a) Schematic illustrations of three competing processes: i. cyclodehydrogenation of two adjacent phenyl groups; ii. C-C bond formation between a phenyl and a methyl group; iii. phenyl group migration. (b) Energy diagram comparing three competing processes. (c) Optimized configurations in (a).

Additionally, we also resorted to NEB calculations for the process (iii) phenyl group migration I-Ia and Ia-II. The calculated results for I-Ia and Ia-II are summarized in Figures R24 and R25. The energy barriers are 1.63 eV and 2.10 eV respectively, indicating the process (iii) phenyl group migration are favoured. We have inserted the Figures R24 and 25 into the Supplementary information as Supplementary Figures 21 and 22 along with the related description into the revised manuscript.

Figure R24. Energy barriers for I to Ia from the NEB calculation.

Figure R25. Energy barriers for Ia to II from the NEB calculation.

Comment: 5) How can the obtainment of specie 3 not arise from direct C-C coupling from 1? The activated methyl group (CH₂*) could directly be attacked by a neighboring phenyl and ultimately lead to specie 3 without implying any methyl migration. Can the author elaborate on that and provide further sup-port for one or the other pathway?

Author reply: We thank the reviewer's suggestion. Experimentally speaking, we observed the high selectivity of product **7** at 440 K, and the rest minor species are also phenyl migrated products but with some defects (loss of phenyl groups). Therefore, the species **3** obtained at

higher temperatures should arise from these migrated species at 440 K. This means that specie **1** indeed undergoes the phenyl migration rather than C-C coupling. From the calculations, the process (iii) phenyl group migration is energetically favorable related to the process (ii) C-C bond formation between a phenyl and a methyl group as discussed above. Therefore, the migration is preferential over C-C coupling.

Comment: 6) Why is the Au(111) surface held at 320 K during deposition of the precursors? Was room temperature somehow not suited?

Author reply: We apologize for the misleading. We gave the substrate temperature for one of our prepared samples. In fact, among a wide range of substrate temperatures, that is, from room temperature to about 440 K, the regular self-assembly stays intact, as shown below in Figure R26. The migration reaction occurs until the temperature reaches about 440 K. We have replaced this temperature with RT in the revised manuscript for accuracy.

Figure R26. DMTPB self-assembly obtained onto Au(111) substrate kept at (a) RT and (b) 320 K and after annealed to (c) 380 K (d) 420 K. It is noted that very rare product **7** is obtained at 420 K, as denoted by a yellow circle in (d).

Comment: 7) Some of the observed reaction products are assigned to chemical structures comprising sp^3 carbon atoms with H atoms pointing away from the substrate (e.g. species **2**, **8**, C1-C4, and others). While this feature is usually well-visible by height-dependent nc-AFM imaging, BR-STM may fail in capturing such subtle detail. Figure S5 reports nc-AFM investigation of specie **8**. However, no clear fingerprint of such feature is observed. Could the authors comment on this?

Author reply: We appreciate the reviewer for this important remark. In previous Figure S5, the nc-AFM image of specie **8** fails to give any fingerprint of the hydrogen atoms at the five-

membered ring apex while this should have been observed. We simulated a series of nc-AFM images of specie **8** comprising hydrogen atoms at the sp³ carbon atoms at different amplitudes of the qPlus sensor as well as applied sample bias, as shown in Figures R27-R29. These results indicate that the products we proposed before do not contain hydrogen atoms at the sp³ carbons.

Figure R27. Simulated nc-AFM images of saturated specie **8** with an amplitude of 100 pm and a bias voltage of 20 mV. Scale: 2*2.5 nm.

Figure R28. Simulated nc-AFM images of saturated specie **8** with an amplitude of 100 pm and a bias voltage of 0 mV. Scale: 2*2.5 nm.

Figure R29. Simulated nc-AFM images of saturated specie **8** with an amplitude of 20 pm and a bias voltage of 0 mV. Scale: 2×2.5 nm.

From Figures R27-R29, the amplitude of the sensor and the applied bias show negligible impact on the acquired nc-AFM images. Therefore, we ruled out any influence of the parameters that may lead to artifacts in our nc-AFM experiments. The observed features are directly dominated by the molecular structure. A comparison of the experimental nc-AFM image and the simulation (Figure R30) indicates the absence of hydrogens at the carbon atom sites of the methyl group in product **8**, as described below in detail:

- (i). In the experimental nc-AFM images obtained at a moderate tip-sample separation, the outer edges (denoted by yellow arrows in Figure R30) of the fluorenes are well resolved, which coincides with the simulated nc-AFM image, while such features are absent in the saturated form in Figure R30c. At a relatively close tip-sample separation, although the edges are also identified for the hydrogen saturated form, the fluorene groups are distorted, which is not in accordance with the experimental results.
- (ii). The apex of the five-membered rings (denoted by the orange arrow in figure R30) are well resolved in the experimental nc-AFM image, which agrees with the simulations. However, it shows a darker feature for the hydrogen saturated form at a moderate tip-sample separation. These dark features originate from the hydrogen atoms at the sp^3 center, and are identified as bright protrusions at larger tip-sample separations.

(iii). The quinoid structure can be directly ruled out due to its complementary planar character, as can be seen in Figure R30g-i.

Figure R30. Three possible chemical structures (a, d, g), optimized adsorption configurations (b, e, h), nc-AFM image (f), and simulated nc-AFM images (c, f, i) of product 8.

We also obtained the ultra-high-resolution BR-STM images of product **14** which does not contain the non-planar phenyl groups, as shown in Figure R31. The absence of any feature at the sites denoted by orange arrows in Figures R31b and 31c indicates hydrogen atoms are not involved. We thus rule out the hydrogen saturated form. The possible canonical structure of **14** by removing the two hydrogen atoms are listed in Figure R31c. The absence of hydrogen atoms is also valid for the other intermediates, as can be seen in the series of high-resolution BR-STM/nc-AFM images shown in the supplementary materials (Supplementary Figures 3, 8, 9, 12, 13, 28).

Figure R31. Structure confirmation for product **14**. (a) Constant current STM image and (b) High-resolution BR-STM image and (c) Zoom-in BR-STM image of the area in (b) marked by an orange square. (d) Three possible structures of product **14**.

At this point, it's noteworthy to furtherly discuss the resonance structure of the cascaded cyclodehydrogenation intermediates. The cyclodehydrogenation intermediates **9**, **14** and **2** can be simply regarded as hydrocarbons with a *p*-quinodimethane core (*p*-QDM) [Acc. Chem. Res. 2017, **50**, 1997-2006; Physical Organic Chemistry of Quinodimethanes, 2018, 107-168. J. Am. Chem. Soc. 2013, **135**, 6363-6371; Nat. Commun. 2021, **12**, 1-7], as shown in Figure R32. Considering that product **2** is largely present at 590 K (27%) as well as it can be found at a higher temperature, which is not in line with its highly reactive diradical character. We thus propose that product **2** may have a closed-shell ground state with four sextets at the corners. This can be also evidenced by a shorter bond length between the fluorene and central benzene ring, as denoted by the yellow arrow in Figure R32c. Similar short bond length is also observed for **9** and **14**, as shown in Figures R32d and R32e. A dominant closed-shell ground state thus is also expected here.

Figure R32. Chemical structures of products **2**, **9**, **14**. (a) Schematic illustration of the possible canonical forms of considered structures. (b, d) STM and (c, e) nc-AFM images of products **2** and **9**, respectively. (f) STM and (g) BR-STM images of product **14**.

As for cascaded cyclodehydrogenation intermediates **12/C4**, **13**, **15**, **16** and **16⁺**, although the additional formed five-membered ring may alter the conjugation, for the sake of simplicity, we here still regard them as indenofluorene derivatives. According to literature reports [J. Phys. Chem. A 2015, **119**, 10620-10627; J. Am. Chem. Soc. 2019, **141**, 12346-12354; Nat. Chem. 2016, **8**, 753-759], indeno[1,2-b]fluorene has a canonical structure with diradical character $y = 0.072$, which means it has a dominant contribution of the closed-shell structure and indeno[2,1-b]fluorene has more open-shell character with $y = 0.645$. Note that product **15** can be also drawn as indeno[2,1-c]fluorene derivative ($y = 0.021$), as shown in Figure R33c, which also has a dominant closed-shell character. Based on the above discussion, we give the dominant resonance structure of these products in Supplementary Figure 11.

Figure R33. Resonance structures of products **12/C4**, **13**, **15**, **16** and **16'**. (a-c) different indenofluorene regioisomers (top) and the corresponding classified products based on the indenofluorene core proposed (bottom).

We assign the fully reacted products of nanographene **3-6** to the closed-shell canonical structure with four sextets as the BR-STM and their corresponding STS do not show any feature near the Fermi level, while the open-shell diradical form often gives rise to detectable features [J. Am. Chem. Soc. 2020, **142**, 1147-1152; Sci. Adv. 2019, **5**, eaav7717. Angew. Chem. Int. Ed. 2021, **60**, 16208-16214]. Besides the STM-based measurements, in the nc-AFM images, the shorter bond length (yellow arrows) and brighter contrast (blue arrows) in Figure R34 indicate a higher bond order of these products, which explains their double bond character.

Figure R34. Chemical structures of products **3**, **4**, **5**, **6**. (a) nc-AFM images of products 3-6 and (b) their chemical structures, respectively.

The hydrogen saturation of products on the Cu(111) substrate is also confirmed. Our simulated nc-AFM images show excellent agreement with the experimental results, as shown in Figure R35. We thus propose that the products on Cu(111) have the quinoid configurations. Other products on the Cu(111) and Ag(110) substrate as shown in the supplementary materials have been modified based on the above discussions.

Figure R35. Chemical structure elucidation of product **C2** on Cu(111). (a) STM image, (b) BR-STM image, (c) optimized adsorption configuration and (d) proposed chemical structure of **C2**, respectively. (e) Experimental and (f-h) simulated nc-AFM images of product **C2** with decreased tip height.

In summary, we proposed that all the experimentally observed products on Au(111), Cu(111) and Ag(110) do not contain the H at sp³ sites. A dominant diradical form or canonical quinoid form is responsible for the observed products. All the mentioned products on the three substrates, i.e., Au(111), Cu(111) and Ag(110), have been now updated according to the above discussion, as shown below in Table R2. We have inserted the corresponding discussion of the resonance form into the supplementary materials as Supplementary Figs. 7-9 and Figs. 14-17 along with the related descriptions.

Table R2: Summarised chemical structures of observed products on Au(111), Cu(111) and Ag(110)

Au(111)						
	Product 2	Product 3	Product 4	Product 5	Product 6	Product 7

							Product 8	Product 9	Product 10	Product 11	Product 12	Product 13
							Product 14	Product 15	Product 16	Product 16	Product 17	Product 18
						
	Product 19					
Cu(111)							Product C1	Product C2	Product C3	Product C4	Product C5	Product C6
						
	Product C7	Product C8	Product C9	Product C10	Product C11	
Ag(110)						
	Product A1	Product A2	Product A3	Product A4	Product A5	

Comment: 8) Figure S5: The model should comprise two H atoms pointing away from the surface, located at the five-membered rings apexes close to the central phenylene. They seem to be absent in the top view image of the optimized chemical model.

Author reply: We appreciate the reviewer very much for pointing this out. Hydrogen atoms play a crucial role in the migration reaction. To provide explicit representation, we have now colored the hydrogen atoms in red of the models throughout the manuscript and the supplementary materials.

Comment: 9) Figure S13: There seems to be confusion with the 1' and 1'' labeling. Does 1A coincide with TS1?

Author reply: We apologize for the caused confusion. 1A indeed coincides with TS1. We have checked all the numbering of the products for consistency in our revised manuscript.

Comment: 10) Page 7, line 129: Product 1c is not the most energetically favored, according to Figure 3f, but rather 1a.

Author reply: We thank the reviewers for pointing this out. We have now updated the results of the calculated energy diagram which is more accurate. Now intermediate 1b (now III) is the most energetically favored, and the interpretation of the mechanism has been also revised according to our updated calculations.

Comment: 11) Some of the discussions reported in the manuscript are hard to read, and a careful revision of the English would be desirable. Moreover, there are some mistakes in the numbering of figures (especially some supplementary figures referred to at page 7) and some typos through the text (e.g. "Methodors") that should be fixed.

Author reply: We apologize for the poor language of our manuscript. We have revised and polished the whole manuscript as well as the supplementary materials. We also update the referred supplementary figures in the revised manuscript to make sure they give the relevant details of the text. Now we believe the revised paper will provide a more readable description of the main results of this study.

REVIEWER COMMENTS

Reviewer #1 (Remarks to the Author):

The authors have done an excellent job in answering the comments of this reviewer and all other reviewers. In my opinion, all critical points have been sufficiently addressed, resulting in a substantially improved manuscript. Publication of this version in Nature Communications is recommended.

Reviewer #2 (Remarks to the Author):

See attached file.

Reviewer #3 (Remarks to the Author):

The authors have satisfactorily addressed all the raised points, and the conclusions appear now more accurate, especially regarding the chemical identification of all species.

However, one critical point now emerged, which deserves careful consideration.

The intermediate 7 is observed after annealing the Au(111) surface to 440K, i.e. the Thiele's hydrocarbon V must form within this temperature. However, the DFT-calculated energy diagram in Figure 3 shows that, to reach the Thiele's hydrocarbon V one must go through high-energy transition states, especially TS1 at 3.14 eV. Such energy looks very high, and based on experience one could argue that annealing Au(111) to 440 K is not enough to overcome this barrier. Can the author comment on this? Can it be that the true reaction path is another one? The absolute energies of the various possible intermediates depicted in Supplementary Fig. 20 as well as the comparison in Supplementary Fig. 27 is not sufficient if one does not compare also the energy barriers in between the steps, i.e. the energy of the corresponding transition states. For instance, in Supplementary Fig. 27 the overall energy barrier to go from I to II is 3.14 eV (see Fig. 3), much higher than I' in path 1. The authors should rationalize this aspect.

Few minor suggestion follow.

The description of the energetics observed in Fig. 3 needs some attention (e.g. lines 144-145). The energy barrier of 2.10 eV adds on top of the energy of the starting state (1.04 eV), leading to a TS1 that is at 3.14 eV.

The numbering of the species in Figure 1 looks confusing. Mixing of Roman and Arabic numbers, and the non-sequential numbering of the species appears odd. Also, numbering before species observed after annealing to 590 K and then species observed at 440 K is misleading. I would suggest to make this clearer.

There should be no space between the element and the surface plane, e.g. Au (111) must be changed into Au(111). Please, check this throughout the entire text.

In the last paragraph before the conclusions, the sentence "The ultra-high yield of uniform

final products (Supplementary Fig. 3j)" should be changed into "The ultra-high yield of the intermediate product 7 (Supplementary Fig. 3j)".

The authors have clarified many things in their manuscript but I still have quite serious remarks about the reaction mechanism calculations. Especially see point 3 below, where I try to motivate why it would be better to remove some of the calculated results from the manuscript. I do not recommend publication of the manuscript in its current form.

1. The new Figure 3 became much clearer. Now that the pathway is better represented, I have a quite major remark about it. Going from I to II there are two barriers that need to be crossed, one of 1.63 eV and one of 2.10 eV. However, there is nothing stopping the system going back from the state Ia to the initial state I. In other words, it is necessary to consider the efficient barrier going from I to II, which is 3.14 eV(?). A quite considerably barrier considering the temperature of 440 K. A similar problem exists to go from II to III. (I think it would be instructive to write out the energy of all states relative the initial state.) I wonder if the pathway being presented is really the lowest energy one. Can the authors really be certain that the reaction is initiated by a hydrogen migration? What is the activation energy the perform a dehydrogenation of the CH₃ group? As correctly stated, once a dehydrogenation has been performed, the reversibility is broken at that point. Anyhow, it would be nice if this claim could be substantiated by how much the entropy of the abstracted hydrogen is expected to contribute to the free energy. In particular, is it enough to make state V thermodynamically favorable compared to the seemingly quite stable state III?

2. The new Figure R16 is quite interesting comparing the energy of different paths depending on when the hydrogen are removed. However, if including entropy contribution from the removed hydrogen, path 2 might become favorable over path 3 and 4. Also, this energy diagram does not say anything about the kinetics of the different possibilities.

3. I am very surprised by the small number of images used in the NEB calculations in Suppl. Figure 21 and 22. If they included more images in these calculations, I am certain they would capture significantly more complex scenarios to get from I to Ia and from Ia to II. For both steps, there are probably more than one barrier considering the large migration of the hydrogen/phenyl group. For the calculations of the transition states of other reaction steps a constrained geometry optimization method was used. How can the authors be sure they locate transition states? Do they perform an analysis of the vibrational eigenmodes? And how can they be sure that each of these steps is associated with only one barrier? The motivation for not performing full NEB calculations for each step is the lack of sufficient computational resources, which is understandable, but I would recommend to remove calculations of reaction barriers, and focus on intermediate steps and present it as preliminary assertion of reaction pathways, and that additional efforts are required to understand the reaction in detail.

I understand the difficulty in considering all possible reaction pathways, in particular in a study where the main point is to introduce novel on-surface reactions, for which one could expect a more comprehensive theoretical follow-up study. At least to me, the new experimental findings are sufficient for publication in Nature Communications, together with a discussion about the possible reaction pathway based in initial calculated results of reaction intermediates.

4. Another issue: In state V, I do not understand why it is necessarily a di-radical. It is possible to draw a resonance structure with only paired electrons (see below). The same holds for product 7.

Point-by-point Response to NCOMMS-21-44910

We thank two reviewers for their constructive comments and suggestions. We have performed extensive additional DFT calculations to discuss the possible reaction pathways. We have addressed all the comments point-by-point and revised the manuscript accordingly. In this response letter, comments from the reviewers are summarized in blue typeface and our detailed responses are in regular black typeface. Our changes to the text are in *italic* typeface.

Reviewer 1:

The authors have done an excellent job in answering the comments of this reviewer and all other reviewers. In my opinion, all critical points have been sufficiently addressed, resulting in a substantially improved manuscript. Publication of this version in Nature Communications is recommended.

Author reply: We thank the reviewer very much for the time devoted to our manuscript, the very high appreciation of our work recommending publication.

Reviewer 2:

The authors have clarified many things in their manuscript but I still have quite serious remarks about the reaction mechanism calculations. Especially see point 3 below, where I try to motivate why it would be better to remove some of the calculated results from the manuscript. I do not recommend publication of the manuscript in its current form.

Author reply: We thank the reviewer very much for the time devoted to our manuscript, and his/her insightful comments, which have been replied point-to-point below, altogether reinforcing the quality of our manuscript.

Comment: 1) The new Figure 3 became much clearer. Now that the pathway is better represented, I have a quite major remark about it. Going from I to II there are two barriers that need to be crossed, one of 1.63 eV and one of 2.10 eV. However, there is nothing stopping the system going back from the state Ia to the initial state I. In other words, it is

necessary to consider the efficient barrier going from I to II, which is 3.14 eV(?). A quite considerably barrier considering the temperature of 440 K. A similar problem exists to go from II to III. (I think it would be instructive to write out the energy of all states relative the initial state.) I wonder if the pathway being presented is really the lowest energy one. Can the authors really be certain that the reaction is initiated by a hydrogen migration? What is the activation energy the perform a dehydrogenation of the CH₃ group? As correctly stated, once a dehydrogenation has been performed, the reversibility is broken at that point. Anyhow, it would be nice if this claim could be substantiated by how much the entropy of the abstracted hydrogen is expected to contribute to the free energy. In particular, is it enough to make state V thermodynamically favorable compared to the seemingly quite stable state III?

Author reply: We thank the reviewer for his/her constructive comments. We followed the reviewer's suggestions and did more calculations with including entropy contribution from the removed hydrogen in the CH₃ group. And we renamed these products according to the reviewer 3's suggestion. The I, II, III, IV and V used before are renamed as 1, 1a, 1b,1c and 1d respectively. The Ia, IIa, *etc.* used before are renamed as int1, int2, *etc.* respectively. As shown in Figures R1-4, energy diagrams with and without including entropy contribution for the stepwise phenyl groups migrations in path 1 (dehydrogenations precede phenyl groups migrations), path 2 (dehydrogenations occur during 1-1a and 1a-1b), path 3 (dehydrogenations occur during 1c-1d and 1d-2) and path 4 (dehydrogenations occur after two phenyl groups migrations) are compared respectively. In addition, since the product 2 is the first product observed after increasing the annealing temperature, we added it in these energy diagrams. As predicted by the reviewer, the entropy contributions from the removed hydrogens indeed cause large changes in energy diagrams. For path 1 (Figure R1), dehydrogenations occur during 1-1' and 1d-2, accompanied with large energy differences between with (dark red) and without (light red) including entropy contribution during 1-1' and 1d-2. For path 2 (Figure R2), dehydrogenations occur during 1-1a, 1a-1b and 1d-2, accompanied with large energy differences between with (dark orange) and without (light orange) including entropy contribution during 1-1a, 1a-1b and 1d-2. For path 3 (Figure R3), dehydrogenations occur during 1b-1c, 1c-1d and 1d-2, accompanied with large energy

differences between with (dark blue) and without (light blue) including entropy contribution during 1b-1c and 1d-2. For path 4 (Figure R4), dehydrogenations occur during 1b-1b' and 1d-2, accompanied with large energy differences between with (dark green) and without (light green) including entropy contribution during 1b-1b' and 1d-2.

Figure R1. (a) Energy diagram with (dark) and without (light) including entropy contribution for the stepwise phenyl groups migrations in path 1: dehydrogenations precede phenyl groups migrations. (b) Schematic illustration of the stepwise phenyl groups migrations from 1 to 2. (c) Optimized configurations in (a). The experimentally observed species 1 and 2 are denoted with solid lines, while the intermediates 1a, 1b, 1c and 1d are denoted with dashed lines.

Figure R2. (a) Energy diagram with (dark) and without (light) including entropy contribution for the stepwise phenyl groups migrations in path 2: dehydrogenations occur during 1-1a and 1a-1b. (b) Schematic illustration of the stepwise phenyl groups migrations from 1 to 1. (c) Optimized configurations in (a). The experimentally observed species 1 and 2 are denoted with solid lines, while the intermediates 1a, 1b, 1c and 1d are denoted with dashed lines.

Figure R3. (a) Energy diagram with (dark) and without (light) including entropy contribution for the stepwise phenyl groups migrations in path 3: dehydrogenations occur during 1b-1c and 1c-1d. (b) Schematic illustration of the stepwise phenyl group migrations from 1 to 1d. (c) Optimized configurations in (a). The experimentally observed species 1 and 2 are denoted with solid lines, while the intermediates 1a, 1b, 1c and 1d are denoted with dashed lines.

Figure R4. (a) Energy diagram with (dark) and without (light) including entropy contribution for the stepwise phenyl groups migrations in path 4: dehydrogenations occur after two phenyl groups migrations. (b) Schematic illustration of the stepwise phenyl groups migrations from 1 to 1d. (c) Optimized configurations in (a). The experimentally observed species 1 and 2 are denoted with solid lines, while the intermediates 1a, 1b, 1c and 1d are denoted with dashed lines.

Energy diagram comparing all four plausible reaction mechanisms with and without including entropy contribution are shown in Figure R5. With including entropy contribution, the energy differences from 1 to product 2 (Figure R5a) become smoother than those without including entropy contribution (Figure R5b). And path 3 (dehydrogenating during 1b-1c and 1c-1d) is the energetic favourable reaction path for both with and without including entropy contribution. For path 3, 1-1a, 1a-1b and 1d-2 are exothermic, while 1b-1c and 1c-1d are endothermic. Since the extra hydrogen atoms are going to get away during 1b-1c and 1c-1d, the reverse reaction conditions are broken, making endothermic reactions 1b-1c and 1c-1d possible to happen. Therefore, even though 1b is more energetic favourable than product 2, the reaction is still going from 1 to product 2 which is the first product observed after increasing the annealing temperature. To make this point clearer, the experimentally observed species 1 and 2 are denoted with solid lines, while the intermediates 1a, 1b, 1c and 1d are denoted with dashed lines.

Figure R5. Energy diagram comparing all four plausible reaction mechanisms (a) with and (b) without including entropy contribution. Path 1: dehydrogenations precede phenyl groups migrations. Path 2: dehydrogenations occur during 1-1a and 1a-1b. Path 3: dehydrogenations occur during 1b-1c and 1c-1d. Path 4: dehydrogenations occur after two phenyl groups migrations. The experimentally observed species 1 and 2 are denoted with solid lines, while the intermediates 1a, 1b, 1c and 1d are denoted with dashed lines.

After ensuring that path 3 (dehydrogenating during 1b-1c and 1c-1d) is the energetic favourable reaction path for both with and without including entropy contribution, we then did more calculations to further understand the phenyl group migration. We appreciate the reviewer for the important remark that the efficient barrier going from 1 to 1a is 3.14 eV since there is nothing stopping the system going back from state 1a to the initial state 1, which is too high to overcome at 440 K. As the reviewer said, we were not sure at first that the reaction is initiated by a hydrogen migration. So, before we chose the hydrogen migration path, two possibilities of phenyl group migration were compared: path A that the reaction is initiated by a hydrogen migration (Figure R6) and path B that the reaction is initiated by a

dehydrogenation (Figure R7). As shown in Figure R8, path A (hydrogen migration) is the energetic favourable reaction path. In the reaction steps 1-int1 (named I-Ia before) and int1-1b (named Ia-II before), we resorted to NEB calculations for each reaction steps. The calculated results show that the energy barriers of 1-int1 and int1-1b are 1.63 eV and 2.16 eV, respectively. However, we ignored that there is nothing stopping the system going back from state int1 to the initial state 1, resulting in a 3.14 eV efficient barrier going from I to II, which is too high to overcome at 440 K. Based on the above discussion, we prefer path B (the reaction is initiated by a dehydrogenation of methyl group, which has the lowest bond dissociation energy [Energy Fuels **2021**, 35, 3, 2224–2233]) which could break the reversibility is the most likely phenyl group migration path, as the reviewer suggested.

We have inserted the Figures R1-5 into the Supplementary information as Supplementary Figures 21-25 along with the related description into the revised manuscript.

Figure R6. (a) Schematic illustration of the stepwise phenyl groups migrations from 1 to 1d in path A: the reaction is initiated by a hydrogen migration. (b) Energy diagram of path A. (c) Optimized configurations in (a). The experimentally observed species 1 and 2 are denoted with solid circles, while the intermediates 1a, 1b, 1c and 1d are denoted with hollow circles.

Figure R7. (a) Schematic illustration of the stepwise phenyl groups migrations from 1 to 1d in path B: the reaction is initiated by a dehydrogenation. (b) Energy diagram of path B. (c) Optimized configurations in (a). The experimentally observed species 1 and 2 are denoted with solid circles, while the intermediates 1a, 1b, 1c and 1d are denoted with hollow circles.

Figure R8. Energy diagram comparing two possibilities of phenyl group migrations. Path A: the reaction is initiated by a hydrogen migration, which schematic illustration and corresponding optimized configurations are in Figure R6. Path B: the reaction is initiated by a dehydrogenation, which schematic illustration and corresponding optimized configurations are in Figure R7. The experimentally observed species 1 and 2 are denoted with solid lines, while the intermediates 1a, 1b, 1c and 1d are denoted with dashed lines.

Comment: 2) The new Figure R16 is quite interesting comparing the energy of different paths depending on when the hydrogen are removed. However, if including entropy contribution from the removed hydrogen, path 2 might become favorable over path 3 and 4. Also, this energy diagram does not say anything about the kinetics of the different possibilities.

Author reply: We thank the reviewer for pointing out this. As shown in Figure R5 in the reply of comment 1, path 3 (dehydrogenating during 1b-1c and 1c-1d) is the energetic favourable reaction path for both with and without including entropy contribution.

Comment: 3) I am very surprised by the small number of images used in the NEB calculations in Suppl. Figure 21 and 22. If they included more images in these calculations, I am certain they would capture significantly more complex scenarios to get from I to Ia and from Ia to II. For both steps, there are probably more than one barrier considering the large migration of the hydrogen/phenyl group. For the calculations of the transition states of other reaction steps a constrained geometry optimization method was used. How can the authors be sure they locate transition states? Do they perform an analysis of the vibrational eigenmodes? And how can they be sure that each of these steps is associated with only one barrier? The motivation for not performing full NEB calculations for each step is the lack of sufficient computational resources, which is understandable, but I would recommend to remove calculations of reaction barriers, and focus on intermediate steps and present it as preliminary assertion of reaction pathways, and that additional efforts are required to understand the reaction in detail.

I understand the difficulty in considering all possible reaction pathways, in particular in a study where the main point is to introduce novel on-surface reactions, for which one could expect a more comprehensive theoretical follow-up study. At least to me, the new experimental findings are sufficient for publication in Nature Communications, together with a discussion about the possible reaction pathway based in initial calculated results of reaction intermediates.

Author reply: We thank the reviewer very much for understanding the difficulty in

performing full NEB calculations for each step. We have performed NEB calculations with more images. Our results showed that there was indeed more than one barrier as the reviewer suspected, resulting that it did not converge for three months. Therefore, we reduced the number of images used in the NEB calculations for both 1-1a and 1a-1b to make it converge faster. The trade-off is that more details are not available. And it is even possible that the reaction path we proposed is not the true reaction path.

We agree with the reviewer that it is better to remove calculations of reaction barriers and present it as preliminary assertion of reaction pathways. Based on the above discussions in the reply of comment 1 that path 3 (Figure R3, dehydrogenating during 1b-1c and 1c-1d) are the most likely reaction paths, we further performed analysis of the vibrational eigenmodes for 1, 1a, 1b, 1c, 1d and 2. There is no imaginary vibrational eigenmode for them, revealing that 1a, 1b, 1c and 1d are all possible intermediate states for 1-2 reaction. We revised Figure 3 (Figure R9) accordingly as below.

Figure R9. (a) Schematic illustration of the stepwise phenyl groups migrations from 1 to 2. (b) Energy diagram with (black) and without (blue) including entropy contribution for the stepwise phenyl groups migrations. (c) Optimized configurations in (a). The experimentally observed species 1 and 2 are denoted with solid circles, while the intermediates 1a, 1b, 1c and

1d are denoted with hollow circles.

We have also inserted the Figure R7 with more detailed of the proposed reaction pathway and the initial calculations into the Supplementary information as Supplementary Figure 20 along with the related description into the revised manuscript.

Comment: 4) Another issue: In state V, I do not understand why it is necessarily a di-radical. It is possible to draw a resonance structure with only paired electrons (see below). The same holds for product 7.

Author reply: We apologize for not making this clearer. In fact, we considered all the possible chemical structures of experimentally observed product 7 and 8 (now referred to as product 2 and 3). Figure R10 shown below gives the hydrogenated form, diradical form and canonical quinoid form. The good agreement between the experimental nc-AFM result and the simulated nc-AFM image of the diradical form indicates that the product 8 indeed hold the diradical form. The nonplanar structure of product 7, as indicated by a white arrow in Figure R11 suggests that product 7 is also in the diradical form rather than the planar quinoid form. Furthermore, we simulated the nc-AFM image of product 7 in the diradical form, which also agrees well with the experimental observations, as shown in Figure R12. The features denoted with coloured arrows are well reproduced. Based on the successful structural elucidation of 7 and 8 (now referred to as product 2 and 3), which are closely related to V (now referred to as 1d), we then also proposed that the intermediate V is also in the diradical form. We also noticed that we didn't observed intermediate V in the experiment, which could be explained by its unstable nature, that is, the easily transformation to 7.

Figure R10. Three possible chemical structures (a, d, g), optimized adsorption configurations (b, e, h), nc-AFM image (f), and simulated nc-AFM images (c, f, i) of product 3 (before referred as product 8).

Figure R11. Experimental (a) STM, (b) nc-AFM and (c) chemical structure of product 2 (before referred as product 7).

Figure R12. (a) Experimental STM, (b) BR-STM and (c-f) nc-AFM images of product **7**. (g) Chemical structures of **7** with diradicals, (h) optimized adsorption configurations and (i-l) a series of simulated nc-AFM images of product **7**.

Reviewer 3:

The authors have satisfactorily addressed all the raised points, and the conclusions appear now more accurate, especially regarding the chemical identification of all species. However, one critical point now emerged, which deserves careful consideration.

Author reply: We thank the reviewer's positive feedback to our work. We have addressed point-to-point his/her comments below, which have contributed to the further improvement of our manuscript.

Comment: 1) The intermediate 7 is observed after annealing the Au(111) surface to 440K, i.e. the Thiele's hydrocarbon V must form within this temperature. However, the DFT calculated energy diagram in Figure 3 shows that, to reach the Thiele's hydrocarbon V one must go through high energy transition states, especially TS1 at 3.14 eV. Such energy looks very high, and based on experience one could argue that annealing Au(111) to 440 K is not enough to overcome this barrier. Can the author comment on this? Can it be that the true reaction path is another one? The absolute energies of the various possible intermediates depicted in Supplementary Fig. 20 as well as the comparison in Supplementary Fig. 27 is not sufficient if one does not compare also the energy barriers in between the steps, i.e. the energy of the corresponding transition states. For instance, in Supplementary Fig. 27 the overall energy barrier to go from I to II is 3.14 eV (see Fig. 3), much higher than I' in path 1. The authors should rationalize this aspect.

Author reply: We appreciate the reviewer for the important remark that the energy barrier going from I to II is 3.14 eV since there is nothing stopping the system going back from state Ia to the initial state I, which is too high to overcome at 440 K. Actually, we are not sure that the proposed path is the true reaction path, since considering all possible reaction pathways are so difficult due to the lack of sufficient computational resources. For examples, we have performed NEB calculations with 11 images for Ia-II. The results show that there was more than one barrier as the reviewer 2 suspected, resulting that it did not converge for three months. Therefore, we reduced the number of images to 6 to make it converge. The trade-off is that more details are not available. And it is even possible that the reaction path we

proposed is not the true reaction path.

As suggested by the reviewer 2 that “I would recommend to remove calculations of reaction barriers, and focus on intermediate steps and present it as preliminary assertion of reaction pathways, and that additional efforts are required to understand the reaction in detail”, we performed more detailed DFT calculations about the intermediate steps and discussed the possible reaction pathways. We renamed these products according to your suggestion. The I, II, III, IV and V used before are renamed as 1, 1a, 1b,1c and 1d respectively. The Ia, IIa, *etc.* used before are renamed as int1, int2, *etc.* respectively.

We did more calculations with including entropy contribution from the removed hydrogen in the CH₃ group. As shown in Figures R13-16, energy diagrams with and without including entropy contribution for the stepwise phenyl groups migrations in path 1 (dehydrogenations precede phenyl groups migrations), path 2 (dehydrogenations occur during 1-1a and 1a-1b), path 3 (dehydrogenations occur during 1c-1d and 1d-2) and path 4 (dehydrogenations occur after two phenyl groups migrations) are compared respectively. In addition, since the product 2 is the first product observed after increasing the annealing temperature, we added it in these energy diagrams. As predicted by the reviewer, the entropy contributions from the removed hydrogens indeed cause large changes in energy diagrams. For path 1 (Figure R13), dehydrogenations occur during 1-1' and 1d-2, accompanied with large energy differences between with (dark red) and without (light red) including entropy contribution during 1-1' and 1d-2. For path 2 (Figure R14), dehydrogenations occur during 1-1a, 1a-1b and 1d-2, accompanied with large energy differences between with (dark orange) and without (light orange) including entropy contribution during 1-1a, 1a-1b and 1d-2. For path 3 (Figure R15), dehydrogenations occur during 1b-1c, 1c-1d and 1d-2, accompanied with large energy differences between with (dark blue) and without (light blue) including entropy contribution during 1b-1c and 1d-2. For path 4 (Figure R16), dehydrogenations occur during 1b-1b' and 1d-2, accompanied with large energy differences between with (dark green) and without (light green) including entropy contribution during 1b-1b' and 1d-2.

Figure R13. (a) Energy diagram with (dark) and without (light) including entropy contribution for the stepwise phenyl groups migrations in path 1: dehydrogenations precede phenyl groups migrations. (b) Schematic illustration of the stepwise phenyl groups migrations from 1 to 2. (c) Optimized configurations in (a). The experimentally observed species 1 and 2 are denoted with solid lines, while the intermediates 1a, 1b, 1c and 1d are denoted with dashed lines.

Figure R14. (a) Energy diagram with (dark) and without (light) including entropy contribution for the stepwise phenyl groups migrations in path 2: dehydrogenations occur during 1-1a and 1a-1b. (b) Schematic illustration of the stepwise phenyl groups migrations from 1 to 1. (c) Optimized configurations in (a). The experimentally observed species 1 and 2 are denoted with solid lines, while the intermediates 1a, 1b, 1c and 1d are denoted with

dashed lines.

Figure R15. (a) Energy diagram with (dark) and without (light) including entropy contribution for the stepwise phenyl groups migrations in path 3: dehydrogenations occur during 1b-1c and 1c-1d. (b) Schematic illustration of the stepwise phenyl groups migrations from 1 to 1d. (c) Optimized configurations in (a). The experimentally observed species 1 and 2 are denoted with solid lines, while the intermediates 1a, 1b, 1c and 1d are denoted with dashed lines.

Figure R16. (a) Energy diagram with (dark) and without (light) including entropy contribution for the stepwise phenyl groups migrations in path 4: dehydrogenations occur after two phenyl groups migrations. (b) Schematic illustration of the stepwise phenyl groups migrations from 1 to 1d. (c) Optimized configurations in (a). The experimentally observed

species 1 and 2 are denoted with solid lines, while the intermediates 1a, 1b, 1c and 1d are denoted with dashed lines.

Energy diagram comparing all four plausible reaction mechanisms with and without including entropy contribution are shown in Figure 17. With including entropy contribution, the energy differences from 1 to product 2 (Figure R17a) become smoother than those without including entropy contribution (Figure R17b). And path 3 (dehydrogenating during 1b-1c and 1c-1d) is the energetic favourable reaction path for both with and without including entropy contribution. For path 3, 1-1a, 1a-1b and 1d-2 are exothermic, while 1b-1c and 1c-1d are endothermic. Since the extra hydrogen atoms are going to get away during 1b-1c and 1c-1d, the reverse reaction conditions are broken, making endothermic reactions 1b-1c and 1c-1d possible to happen. Therefore, even though 1b is more energetic favourable than product 2, the reaction is still going from 1 to product 2 which is the first product observed after increasing the annealing temperature. To make this point clearer, the experimentally observed species 1 and 2 are denoted with solid lines, while the intermediates 1a, 1b, 1c and 1d are denoted with dashed lines.

Figure R17. Energy diagram comparing all four plausible reaction mechanisms (a) with and (b) without including entropy contribution. Path 1: dehydrogenations precede phenyl groups migrations. Path 2: dehydrogenations occur during 1-1a and 1a-1b. Path 3: dehydrogenations occur during 1b-1c and 1c-1d. Path 4: dehydrogenations occur after two phenyl groups migrations. The experimentally observed species 1 and 2 are denoted with solid lines, while the intermediates 1a, 1b, 1c and 1d are denoted with dashed lines.

After ensuring that path 3 (dehydrogenating during 1b-1c and 1c-1d) is the energetic favourable reaction path for both with and without including entropy contribution, we then did more calculations to further understand the phenyl group migration. We appreciate the reviewer for the important remark that the efficient barrier going from 1 to 1a is 3.14 eV since there is nothing stopping the system going back from state 1a to the initial state 1, which is too high to overcome at 440 K. We were not sure at first that the reaction is initiated by a hydrogen migration. So, before we chose the hydrogen migration path, two possibilities of phenyl group migration were compared: path A that the reaction is initiated by a hydrogen migration (Figure R18) and path B that the reaction is initiated by a dehydrogenation (Figure R19). As shown in Figure R20, path A (hydrogen migration) is the energetic favourable reaction path. In the reaction steps 1-int1 (named I-Ia before) and int1-1b (named Ia-II before), we resorted to NEB calculations for each reaction steps. The calculated results show that the energy barriers of 1-int1 and int1-1b are 1.63 eV and 2.16 eV, respectively. However, we ignored that there is nothing stopping the system going back from state int1 to the initial state 1, resulting in a 3.14 eV efficient barrier going from I to II, which is too high to overcome at 440 K. Based on the above discussion, we prefer path B (the reaction is initiated by a dehydrogenation of methyl group, which has the lowest bond dissociation energy [Energy Fuels **2021**, 35, 3, 2224–2233]) which could break the reversibility is the most likely phenyl group migration path.

Figure R18. (a) Schematic illustration of the stepwise phenyl groups migrations from 1 to 1d in path A: the reaction is initiated by a hydrogen migration. (b) Energy diagram of path A. (c) Optimized configurations in (a). The experimentally observed species 1 and 2 are denoted with solid circles, while the intermediates 1a, 1b, 1c and 1d are denoted with hollow circles.

Figure R19. (a) Schematic illustration of the stepwise phenyl groups migrations from 1 to 1d in path B: the reaction is initiated by a dehydrogenation. (b) Energy diagram of path B. (c) Optimized configurations in (a). The experimentally observed species 1 and 2 are denoted with solid circles, while the intermediates 1a, 1b, 1c and 1d are denoted with hollow circles.

Figure R20. Energy diagram comparing two possibilities of phenyl group migrations. Path A: the reaction is initiated by a hydrogen migration, which schematic illustration and corresponding optimized configurations are in Figure R6. Path B: the reaction is initiated by a dehydrogenation, which schematic illustration and corresponding optimized configurations are in Figure R7. The experimentally observed species 1 and 2 are denoted with solid lines, while the intermediates 1a, 1b, 1c and 1d are denoted with dashed lines.

Based on the above discussions, we revised Figure 3 in the maintext accordingly (Figure R21). We also have inserted the Figures R13-17 and Figure R19 into the Supplementary information as Supplementary Figures 21-25 and Supplementary Figure 20 along with the related description into the revised manuscript.

Figure R21. (a) Schematic illustration of the stepwise phenyl groups migrations from 1 to 2.

(b) Energy diagram with (dark) and without (light) including entropy contribution for the stepwise phenyl groups migrations. (c) Optimized configurations in (a). The experimentally observed species 1 and 2 are denoted with solid circles, while the intermediates 1a, 1b, 1c and 1d are denoted with hollow circles.

Comment: 2) Few minor suggestions follow.

The description of the energetics observed in Fig. 3 needs some attention (e.g. lines 144-145). The energy barrier of 2.10 eV adds on top of the energy of the starting state (1.04 eV), leading to a TS1 that is at 3.14 eV.

Author reply: We thank the reviewer for pointing out this important aspect. We have now revised the Figure 3 in the manuscript and the related description of the calculation results. All the barriers now have been removed.

Comment: 3) The numbering of the species in Figure 1 looks confusing. Mixing of Roman and Arabic numbers, and the non-sequential numbering of the species appears odd. Also, numbering before species observed after annealing to 590 K and then species observed at 440 K is misleading. I would suggest to make this clearer.

Author reply: We apologize for making this confusing. Now in our revised manuscript, we use more unified numbering of all the experimental products, the chemical structures used in calculations.

On Au(111), we can obtain different products upon annealing at different temperatures. To make Figure 1 more concise, we renumber the final products at 590 K as A1, A2, A3... and for the intermediates with partially dehydrogenated, we renumber them with Arabic numbers 1(precursor), 2, 3...

For products on Cu(111) and Ag(110) which serve as a justification of the migration reaction on Au(111), we simply use the B1, B2 and C1, C2 for the numbering for all the products. We have revised all the number in the imaged involved in our revised manuscript.

Comment: 4) There should be no space between the element and the surface plane, e.g. Au (111) must be changed into Au(111). Please, check this throughout the entire text.

Author reply: We apologize for making this mistake. We have removed the space between the element and the surface plane in our revised manuscript.

Comment: 5) In the last paragraph before the conclusions, the sentence "The ultra-high yield of uniform final products (Supplementary Fig. 3j)" should be changed into "The ultra-high yield of the intermediate product 7 (Supplementary Fig. 3j)".

Author reply: We thank the reviewer for this important mark. We have changed this sentence into "The ultra-high yield of the intermediate product 2 (Supplementary Fig. 3j) " in our revised manuscript.

REVIEWER COMMENTS

Reviewer #2 (Remarks to the Author):

The authors have done a thorough job revising their manuscript. I have one remaining comment. As I wrote in a previous report, the experiments presented in the study is sufficient for publication in Nature Communications. And I appreciate that the authors removed calculations of barriers, as these may end up being misleading rather than instructive. I believe the manuscript can be recommended for publication following clarifications of some of the new results.

There are some uncertainties I'd like to authors to clarify regarding the calculations. Specifically for the calculations taking entropy into account. Exactly how was this done? This needs to be specified in the Supporting Information. In fact, the authors should be specific what is included both for calculations with and without entropy (I guess that the calculations without entropy only includes electronic enthalpy at 0 K?) I am a bit confused since the differences with and without entropy seems to only depend on when hydrogen atoms are removed from the system in Figure R1, while there is a difference between steps no including hydrogen abstraction in Figure R9. Some clarifications are needed.

Reviewer #3 (Remarks to the Author):

The new changes made by the authors have significantly improved the manuscript and lifted the main criticisms raised in the previous round of revisions. In my view, the text would still benefit from a polishing and rephrasing of some heavy sections, but the scientific statements look now more accurate.

Few minor point to be addressed before final acceptance:

At the end of the introduction, only products A1-A4 are mentioned, while A5 is missing. The same applies to the first line of the caption of Figure 1.

The labels of the various species are wrongly reported in Figure 2, compared to all other figures. I.e., the specie in Figure 2d,i should be A1 and not A2. This should be checked and fixed throughout the text.

Product A1 cannot arise from specie 2 unless one assumes a C-C bond breaking at the five-membered ring site. It could be formed directly from 1d (i.e. without undergoing through the formation of 2), but this would conflict with the reported yields of 95% for 2 at 440 K and 42% for A1 at 590 K. Therefore, I would suggest to add a sentence that clarifies this aspect and discusses the possible C-C bond opening that would justify the formation of A1 from 2.

Point-by-point Response to NCOMMS-21-44910B

We thank two reviewers for their constructive comments and suggestions. We have provided more details of the calculations we performed and addressed all the comments point-by-point and revised the manuscript accordingly. In this response letter, comments from the reviewers are summarized in blue typeface and our detailed responses are in regular black typeface. Our changes to the text are in *italic* typeface.

Reviewer 2:

There are some uncertainties I'd like to authors to clarify regarding the calculations. Specifically for the calculations taking entropy into account. Exactly how was this done? This needs to be specified in the Supporting Information. In fact, the authors should be specific what is included both for calculations with and without entropy (I guess that the calculations without entropy only includes electronic enthalpy at 0 K?) I am a bit confused since the differences with and without entropy seems to only depend on when hydrogen atoms are removed from the system in Figure R1, while there is a difference between steps no including hydrogen abstraction in Figure R9. Some clarifications are needed.

Author reply: We thank the reviewer very much for the time devoted to our manuscript and for pointing out this. As suggested by the reviewer, we provided more calculation details in the Supplementary Information as follows:

“In this paper, we analyzed the reaction process of DMTPB molecules on the Au(111) surface *via* the DFT total energies (enthalpic and entropic effects excluded) and Gibbs free energies (enthalpic and entropic effects included) associated with its adsorption. The DFT total energies obtained in VASP calculations are only electronic energies, which do not include some contributions from the internal energy, enthalpy, and entropy. The Gibbs free energies account for enthalpic and entropic effects at realistic temperatures when evaluating the favorability of molecular adsorption on the Au(111) surface. Therefore, we need to calculate the additional terms contributing to the Gibbs

free energy.

Firstly, the DFT electronic energies do not account for the zero point energy (ZPE) arising from atomic vibrations. The ZPE correction must be added to the electronic energy to calculate the internal energy. The total ZPE correction will be the sum of the individual energies arising from each vibrational mode, K , where there are $3n-5$ vibrational modes in an n -atom linear molecule, $3n-6$ in a nonlinear molecule, and $3n-3$ in a crystal. The internal energy is calculated as:

$$E = E_{\text{elec}} + \text{ZPE} = E_{\text{elec}} + \sum_K \frac{1}{2} h\nu_K \quad (1)$$

where the final summation is the total ZPE correction.

At absolute zero and zero pressure, the internal energy and the enthalpy are equal. To account for enthalpy corrections at nonzero temperatures, the heat capacity must be integrated at constant pressure from absolute zero to the temperature of interest, so that the enthalpy is defined as:

$$H = E + \int_0^T C_p(T') dT' \quad (2)$$

Finally, the Gibbs free energy can be calculated by subtracting the contribution from entropy:

$$G = H - TS \quad (3)$$

Obtaining the Gibbs free energy completely from first principles, therefore, requires the calculation of the ZPE correction, the enthalpic temperature correction, and the entropic correction:

$$G = E_{\text{elec}} + \text{ZPE} + \int_0^T C_p(T') dT' - TS \quad (4)$$

When calculating the free energies of the clean slab and the slab with the adsorbed molecules, the vibrational component is the only contribution that must be considered, since there are no longer any rotational or translational contributions to the free energy corrections. The total entropy is:

$$S = k_B \left[\ln(q_v) + T \frac{\partial \ln(q_v)}{\partial T} + 1 \right] = k_B \left[\sum_K \ln \left(\frac{1}{1 - e^{-\theta_{v,k}/T}} \right) + \sum_K \frac{\theta_{v,k}}{T} \left(\frac{1}{e^{\theta_{v,k}/T} - 1} + 1 \right) \right] \quad (5)$$

and the total enthalpic temperature correction is:

$$H_v(T) - H_v(0) = k_B T \left[\frac{T}{q_v} \frac{\partial q_v}{\partial T} + 1 \right] = k_B T \left[\sum_K \frac{\theta_{v,k}}{T} \left(\frac{1}{e^{\theta_{v,k}/T} - 1} + 1 \right) \right] \quad (6)$$

Zero-point energy (ZPE), enthalpy, and entropy contributions to free energies of DMTPB molecules on the Au(111) surface at 440 K were calculated from vibrational modes of surface species, which were computed with the finite difference approach as implemented in the VaspGibbs code [J. Phys. Chem. C 2013, 117, 26048–26059]. The ZPE, entropic, and enthalpic contributions for the adsorbed slabs in Paths 1-4 at 440 K are listed in Supplementary Tables 2-5.

Supplementary Table 2. ZPE, entropic, enthalpic temperature contributions and relative Gibbs free energies for the adsorbed slabs in Path 1 at 440 K. The DFT total energy E_{elec} and Gibbs free energy G of the experimentally observed species 1 are set as zero points respectively.

	ZPE (eV)	S (eV/K)	$-TS$ (eV)	$\int_0^T C_p(T')dT'$ (eV)	$ZPE+$ $\int_0^T C_p(T')dT'-TS$ (eV)	E_{elec} (eV)	$E_{elec}+ZPE+$ $\int_0^T C_p(T')dT'-TS$ (eV)	G (eV)
1	13.72	5.22×10^{-2}	-22.95	8.62	-0.61	0	-0.61	0
1'	13.10	5.20×10^{-2}	-22.87	8.59	-1.17	2.10	0.93	1.54
1a	13.11	5.21×10^{-2}	-22.91	8.60	-1.20	1.20	0	0.61
1b	13.15	5.16×10^{-2}	-22.71	8.55	-1.00	0.27	-0.73	-0.12
1c	13.16	5.20×10^{-2}	-22.88	8.58	-1.06	0.62	-0.44	0.17
1d	13.18	5.19×10^{-2}	-22.84	8.57	-1.04	0.82	-0.22	0.39
2	12.58	5.14×10^{-2}	-22.62	8.49	-1.53	0.68	-0.85	-0.24

Supplementary Table 3. ZPE, entropic, enthalpic temperature contributions and relative Gibbs free energies for the adsorbed slabs in Path 2 at 440 K. The DFT total energy E_{elec} and Gibbs free energy G of the experimentally observed species 1 are set as zero points respectively.

	ZPE (eV)	S (eV/K)	$-TS$ (eV)	$\int_0^T C_p(T')dT'$ (eV)	$ZPE+$ $\int_0^T C_p(T')dT'-TS$ (eV)	E_{elec} (eV)	$E_{elec}+ZPE+$ $\int_0^T C_p(T')dT'-TS$ (eV)	G (eV)
1	13.72	5.22×10^{-2}	-22.95	8.62	-0.61	0	-0.61	0
1a	13.43	5.187×10^{-2}	-22.83	8.564	-0.83	-0.26	-1.09	-0.48

1b	13.15	5.16×10^{-2}	-22.71	8.55	-1.00	0.27	-0.73	-0.12
1c	13.16	5.20×10^{-2}	-22.88	8.58	-1.06	0.62	-0.44	0.17
1d	13.18	5.19×10^{-2}	-22.84	8.57	-1.04	0.82	-0.22	0.39
2	12.58	5.14×10^{-2}	-22.62	8.49	-1.53	0.68	-0.85	-0.24

Supplementary Table 4. ZPE, entropic, enthalpic temperature contributions and relative Gibbs free energies for the adsorbed slabs in Path 3 at 440 K. The DFT total energy E_{elec} and Gibbs free energy G of the experimentally observed species 1 are set as zero points respectively.

	ZPE (eV)	S (eV/K)	$-TS$ (eV)	$\int_0^T C_p(T')dT'$ (eV)	ZPE+ $\int_0^T C_p(T')dT'-TS$ (eV)	E_{elec} (eV)	$E_{elec}+ZPE+$ $\int_0^T C_p(T')dT'-TS$ (eV)	G (eV)
1	13.72	5.22×10^{-2}	-22.95	8.62	-0.61	0	-0.61	0
1a	13.77	5.20×10^{-2}	-22.87	8.61	-0.49	-0.77	-1.26	-0.65
1b	13.76	5.14×10^{-2}	-22.63	8.54	-0.33	-1.26	-1.59	-0.98
1c	13.43	5.20×10^{-2}	-22.88	8.57	-0.90	-0.04	-0.94	-0.33
1d	13.18	5.19×10^{-2}	-22.84	8.57	-1.04	0.82	-0.22	0.39
2	12.58	5.14×10^{-2}	-22.62	8.49	-1.53	0.68	-0.85	-0.24

Supplementary Table 5. ZPE, entropic, enthalpic temperature contributions and relative Gibbs free energies for the adsorbed slabs in Path 4 at 440 K. The DFT total energy E_{elec} and Gibbs free energy G of the experimentally observed species 1 are set as zero points respectively.

	ZPE (eV)	S (eV/K)	$-TS$ (eV)	$\int_0^T C_p(T')dT'$ (eV)	ZPE+ $\int_0^T C_p(T')dT'-TS$ (eV)	E_{elec} (eV)	$E_{elec}+ZPE+$ $\int_0^T C_p(T')dT'-TS$ (eV)	G (eV)
1	13.72	5.22×10^{-2}	-22.95	8.62	-0.61	0	-0.61	0
1a	13.77	5.20×10^{-2}	-22.87	8.61	-0.49	-0.77	-1.26	-0.65
1b	13.76	5.14×10^{-2}	-22.63	8.54	-0.33	-1.26	-1.59	-0.98
1b'	13.13	5.17×10^{-2}	-22.76	8.55	-1.07	0.19	-0.88	-0.27

1c	13.16	5.20×10^{-2}	-22.88	8.58	-1.06	0.62	-0.44	0.17
1d	13.18	5.19×10^{-2}	-22.84	8.57	-1.04	0.82	-0.22	0.39
2	12.58	5.14×10^{-2}	-22.62	8.49	-1.53	0.68	-0.85	-0.24

Accordingly, energy diagrams with and without enthalpic and entropic effects for the stepwise phenyl groups migrations in path 1 (dehydrogenations precede phenyl groups migrations), path 2 (dehydrogenations occur during 1-1a and 1a-1b), path 3 (dehydrogenations occur during 1b-1c and 1c-1d) and path 4 (dehydrogenations occur after two phenyl groups migrations) are compared respectively, as shown in Supplementary Figures 21-24.

Supplementary Figure 21. Energy diagram of the potential stepwise migration reaction (path 1). (a) Energy diagram with (dark) and without (light) including enthalpic and entropic effects for the stepwise phenyl groups migrations in path 1: dehydrogenations precede phenyl groups migrations. (b) Schematic illustration of the stepwise phenyl group migrations from 1 to 2. (c) Optimized configurations in (a). The experimentally observed species 1 and 2 are denoted with solid lines, while the intermediates 1a, 1b, 1c and 1d are denoted with dashed lines.

Supplementary Figure 22. Energy diagram of the potential stepwise migration reaction (path 2). (a) Energy diagram with (dark) and without (light) including enthalpic and entropic effects for the stepwise phenyl groups migrations in path 2: dehydrogenations occur during 1-1a and 1a-1b. (b) Schematic illustration of the stepwise phenyl group migrations from 1 to 1. (c) Optimized configurations in (a). The experimentally observed species 1 and 2 are denoted with solid lines, while the intermediates 1a, 1b, 1c and 1d are denoted with dashed lines.

Supplementary Figure 23. Energy diagram of the potential stepwise migration reaction (path 3). (a) Energy diagram with (dark) and without (light) including enthalpic and entropic effects for the stepwise phenyl groups migrations in path 3: dehydrogenations occur during 1b-1c and 1c-1d. (b) Schematic illustration of the

stepwise phenyl group migrations from 1 to 1d. (c) Optimized configurations in (a). The experimentally observed species 1 and 2 are denoted with solid lines, while the intermediates 1a, 1b, 1c and 1d are denoted with dashed lines.

Supplementary Figure 24. Energy diagram of the potential stepwise migration reaction (path 4). (a) Energy diagram with (dark) and without (light) including enthalpic and entropic effects for the stepwise phenyl groups migrations in path 4: dehydrogenations occur after two phenyl groups migrations. (b) Schematic illustration of the stepwise phenyl group migrations from 1 to 1d. (c) Optimized configurations in (a). The experimentally observed species 1 and 2 are denoted with solid lines, while the intermediates 1a, 1b, 1c and 1d are denoted with dashed lines.”

We have also added the following discussion into the method section in the revised manuscript:

“Zero-point energy (ZPE), enthalpy, and entropy contributions to free energies of DMTPB molecules on the Au(111) surface at 440 K were calculated from vibrational modes of surface species, which were computed with the finite difference approach as implemented in the VaspGibbs code.⁵¹”

For the comment that “I am a bit confused since the differences with and without entropy seems to only depend on when hydrogen atoms are removed from the system in Figure R1, while there is a difference between steps no including hydrogen abstraction in Figure R9. Some clarifications are needed.” Figures R1 and R9 in the previous version of the Point-by-point response correspond to potential stepwise

migration reaction paths 1 (Supplementary Table 2 and Supplementary Figure 21) and 3 (Supplementary Table 4, Supplementary Figure 23 and Figure 3), respectively. As listed in Supplementary Table 2 or shown in Supplementary Figure 21, the DFT total energy differences ΔE_{elec} /Gibbs free energy differences ΔG of 1-1'(-2×H), 1'-1a, 1a-1b, 1b-1c, 1c-1d, and 1d-2(-2×H) in Path 1 are 2.10/1.54(-2×H), -0.90/-0.93, -0.93/-0.73, 0.35/0.29, 0.20/0.22, -0.14/-0.63(-2 × H) eV respectively. As listed in Supplementary Table 4 or shown in Supplementary Figure 23, the DFT total energy differences ΔE_{elec} /Gibbs free energy differences ΔG of 1-1a, 1a-1b, 1b-1c(-H), 1c-1d(-H), and 1d-2(-2×H) in Path 1 are -0.77/-0.65, -0.49/-0.33, 1.22/0.65(-H), 0.86/0.72(-H), and -0.14/-0.63(-2×H) eV respectively. There are energy differences between the DFT total energies (enthalpic and entropic effects excluded) and Gibbs free energies (enthalpic and entropic effects included) for each step. And when hydrogen atoms are removed from the system, the energy differences are relatively large. To make this point clearer, we added the following discussion in the Supplementary Information:

“As listed in Supplementary Tables 2-5 or shown in Supplementary Figures 21-25, there are energy differences between the DFT total energies (enthalpic and entropic effects excluded) and Gibbs free energies (enthalpic and entropic effects included) for each step. And when hydrogen atoms are removed from the system, the energy differences are relatively large.”

Reviewer 3:

The new changes made by the authors have significantly improved the manuscript and lifted the main criticisms raised in the previous round of revisions. In my view, the text would still benefit from a polishing and rephrasing of some heavy sections, but the scientific statements look now more accurate.

Author reply: We thank the reviewer very much for the time devoted to our manuscript and his/her positive feedback.

Comment: 1) Few minor points to be addressed before final acceptance:

At the end of the introduction, only products A1-A4 are mentioned, while A5 is missing. The same applies to the first line of the caption of Figure 1.

Author reply: We thank the reviewer for pointing this out. We have added the conjugated product A5 which coexists with A1-A4 into the introduction as well as the caption of Figure 1.

Comment: 2) The labels of the various species are wrongly reported in Figure 2, compared to all other figures. I.e., the specie in Figure 2d,i should be A1 and not A2. This should be checked and fixed throughout the text.

Author reply: We thank the reviewer for this important mark. We have changed the labels of the various species in Figure 2 for consistency, together with a thorough check of all the labeling throughout the manuscript and the supplementary information.

Comment: 3) Product A1 cannot arise from specie 2 unless one assumes a C-C bond breaking at the five-membered ring site. It could be formed directly from 1d (i.e. without undergoing through the formation of 2), but this would conflict with the reported yields of 95% for 2 at 440 K and 42% for A1 at 590 K. Therefore, I would suggest to add a sentence that clarifies this aspect and discusses the possible C-C bond opening that would justify the formation of A1 from 2.

Author reply: We thank the reviewer for pointing out this very important aspect. The possible causes of this conflict that specie **2** has an ultrahigh yield at 440 K while specie **A1** dominates at 590 K could be: i) the C-C bond opening at the five-membered ring site, as suggested by the reviewer. This is the most possible case since we also observe species **8**, **9** and **11** at slightly higher temperatures, which cannot be obtained by a simple ring closure reaction from **2**. ii) the desorption of different species during the annealing process could also lead to an increased proportion of **A1** at 590 K since specie **4** (very low yield at 440 K) can transform to **A1**. To make this more clearly, we have inserted the following sentence into the maintext (lines 115-118), as listed below:

*It's also noticed that product **A1** with a high yield at 590 K cannot be obtained directly from the dominating intermediate **2** at 440 K, which suggested a possible ring opening of intermediate **2** at higher temperatures (see Supplementary Tab. 1 and its related discussion for more details).*

To expound on this, we also insert the following discussion into the Supplementary Information (below Supplementary Tab. 1), as listed below:

*The high yield of **A1** at 590 K suggested a structural rearrangement of intermediate **2** at higher temperatures since it cannot be formed directly from the dominating intermediate **2**. Considering the fact that species **8**, **9** and **11** (Supplementary Fig. 10) are also observed at a slightly higher temperature (445 K-465 K), a C-C bond cleavage between the two migrated phenyl groups and subsequent ring closure between the phenyl groups and the central benzene ring is expected and accounts for the unusually high yield of **A1**. Additionally, **A1** can be obtained from specie **4**, and the desorption of other species during the annealing process could potentially lead to an increased proportion of **A1** at 590 K. However, the relatively low yield of **4** would not result in such a high yield of **A1** at 490 K, a ring opening of the fluorene group thus prevails.*

REVIEWERS' COMMENTS

Reviewer #2 (Remarks to the Author):

I appreciate the details added about the calculations. There are still a couple of uncertainties. First of all, the authors add an enthalpy correction by integrating the heat capacity. But they never specify which heat capacity this is.

Furthermore, following each dehydrogenation step, the ZPE drops significantly. I guess this is due to the fewer degrees of freedom for dehydrogenated molecules. But what is not clear is how the entropy and enthalpy of the removed hydrogen are treated within this free energy picture. Maybe it is all there, and just need to be described more precisely.

Reviewer #3 (Remarks to the Author):

The final changes to the manuscript made by the authors satisfy the requests from the last review iteration, and the manuscript looks now ready to be accepted for publication in Nature Communications.

Point-by-point Response to NCOMMS-21-44910C

We thank two reviewers for their constructive comments and suggestions. We have provided more details of the calculations we performed and addressed all the comments point-by-point and revised the manuscript accordingly. In this response letter, comments from the reviewers are summarized in blue typeface and our detailed responses are in regular black typeface. Our changes to the text are in *italic* typeface.

Reviewer 2:

1. I appreciate the details added about the calculations. There are still a couple of uncertainties. First of all, the authors add an enthalpy correction by integrating the heat capacity. But they never specify which heat capacity this is.

Author reply: We thank the reviewer very much for the time devoted to our manuscript and his/her insightful comments. The thermal corrections to the enthalpy can be expressed as a function of the molecular partition function, where the correction from absolute zero to a specified temperature involves integrating the heat capacity at constant pressure:

$$H(T) - H(0) = \int_0^T C_p(T')dT' = k_B T^2 \frac{\partial \ln(q)}{\partial T} + k_B T$$

To make the constant pressure heat capacity clearer, the related discussion in Supplementary Information is revised as follows:

“At absolute zero and zero pressure, the internal energy and the enthalpy are equal. To account for enthalpy corrections at nonzero temperatures, the thermal corrections to the enthalpy can also be expressed as a function of the molecular partition function, where the correction from absolute zero to a specified temperature involves integrating the heat capacity at constant pressure:

$$H(T) - H(0) = \int_0^T C_p(T')dT' = k_B T^2 \frac{\partial \ln(q)}{\partial T} + k_B T \quad (2)”$$

2. Furthermore, following each dehydrogenation step, the ZPE drops significantly. I guess this is due to the fewer degrees of freedom for dehydrogenated molecules. But what is not clear is how the entropy and enthalpy of the removed hydrogen are treated within this free energy picture. Maybe it is all there, and just need to be described more precisely.

Author reply: We thank the reviewer for bringing up this important point. At 440 K where the reactions take place, the removed hydrogen atoms will combine and desorb as H₂. Therefore, the energy of the removed hydrogen atom can be treated as $\frac{1}{2}E(\text{H}_2)$. We are very sorry that the thermal corrections of the removed hydrogen were ignored before. We further calculated the Gibbs free energy of H₂ molecule at 440 K which Zero-point energy (ZPE), enthalpy, and entropy contributions are included, via computing with the finite difference approach as implemented in the VaspGibbs code. The energy difference between the Gibbs free energy of H₂ and the DFT total energy of H₂ is 0.10 eV. The thermal correction $\Delta E(\text{H})$ of the removed hydrogen can be treated as 0.05 eV per H atom. the related discussion in Supplementary Information is added as follows:

“At 440 K where the reactions take place, the removed hydrogen atoms will combine and desorb as H₂. Therefore, the energy of the removed hydrogen atom can be treated as $\frac{1}{2}E(\text{H}_2)$. The Gibbs free energies of H₂ molecules at 440 K which Zero-point energy (ZPE), enthalpy, and entropy contributions are included were computed with the finite difference approach as implemented in the VaspGibbs code. The energy difference between the Gibbs free energy of H₂ and the DFT total energy of H₂ is 0.10 eV. The thermal correction $\Delta E(\text{H})$ of the removed hydrogen can be treated as 0.05 eV per H atom.”

Supplementary Tables 2-5 and Supplementary Figures 21-24 are revised as follows:

Supplementary Table 2. ZPE, entropic, enthalpic temperature contributions and relative Gibbs free energies for the adsorbed slabs in Path 1 at 440 K. The DFT total energy E_{elec} and Gibbs free energy G of the experimentally observed species 1 are set

as zero points respectively.

	ZPE (eV)	S (eV/K)	-TS (eV)	$\int_0^T C_p(T')dT'$ (eV)	ZPE+ $\int_0^T C_p(T')dT'-TS$ (eV)	E_{elec} (eV)	$E_{elec}+ZPE+\int_0^T C_p(T')dT'-TS+\frac{n}{2}\Delta E(H)$ (eV)	G (eV)
1	13.72	5.22×10^{-2}	-22.95	8.62	-0.61	0	-0.61	0
1'	13.10	5.20×10^{-2}	-22.87	8.59	-1.17	2.10	1.03	1.64
1a	13.11	5.21×10^{-2}	-22.91	8.60	-1.20	1.20	0.1	0.71
1b	13.15	5.16×10^{-2}	-22.71	8.55	-1.00	0.27	-0.63	-0.02
1c	13.16	5.20×10^{-2}	-22.88	8.58	-1.06	0.62	-0.34	0.27
1d	13.18	5.19×10^{-2}	-22.84	8.57	-1.04	0.82	-0.12	0.49
2	12.58	5.14×10^{-2}	-22.62	8.49	-1.53	0.68	-0.75	-0.14

Supplementary Table 3. ZPE, entropic, enthalpic temperature contributions and relative Gibbs free energies for the adsorbed slabs in Path 2 at 440 K. The DFT total energy E_{elec} and Gibbs free energy G of the experimentally observed species 1 are set as zero points respectively.

	ZPE (eV)	S (eV/K)	-TS (eV)	$\int_0^T C_p(T')dT'$ (eV)	ZPE+ $\int_0^T C_p(T')dT'-TS$ (eV)	E_{elec} (eV)	$E_{elec}+ZPE+\int_0^T C_p(T')dT'-TS+\frac{n}{2}\Delta E(H)$ (eV)	G (eV)
1	13.72	5.22×10^{-2}	-22.95	8.62	-0.61	0	-0.61	0
1a	13.43	5.187×10^{-2}	-22.83	8.564	-0.83	-0.26	-1.04	-0.43
1b	13.15	5.16×10^{-2}	-22.71	8.55	-1.00	0.27	-0.63	-0.02
1c	13.16	5.20×10^{-2}	-22.88	8.58	-1.06	0.62	-0.34	0.27
1d	13.18	5.19×10^{-2}	-22.84	8.57	-1.04	0.82	-0.12	0.49
2	12.58	5.14×10^{-2}	-22.62	8.49	-1.53	0.68	-0.75	-0.14

Supplementary Table 4. ZPE, entropic, enthalpic temperature contributions and relative Gibbs free energies for the adsorbed slabs in Path 3 at 440 K. The DFT total energy E_{elec} and Gibbs free energy G of the experimentally observed species 1 are set as zero points respectively.

	ZPE (eV)	S (eV/K)	$-TS$ (eV)	$\int_0^T C_p(T')dT'$ (eV)	$ZPE+$ $\int_0^T C_p(T')dT'-TS$ (eV)	E_{elec} (eV)	$E_{elec}+ZPE+$ $\int_0^T C_p(T')dT'-$ $TS+\frac{n}{2}\Delta E(H)$ (eV)	G (eV)
1	13.72	5.22×10^{-2}	-22.95	8.62	-0.61	0	-0.61	0
1a	13.77	5.20×10^{-2}	-22.87	8.61	-0.49	-0.77	-1.26	-0.65
1b	13.76	5.14×10^{-2}	-22.63	8.54	-0.33	-1.26	-1.59	-0.98
1c	13.43	5.20×10^{-2}	-22.88	8.57	-0.90	-0.04	-0.89	-0.28
1d	13.18	5.19×10^{-2}	-22.84	8.57	-1.04	0.82	-0.12	0.49
2	12.58	5.14×10^{-2}	-22.62	8.49	-1.53	0.68	-0.75	-0.14

Supplementary Table 5. ZPE, entropic, enthalpic temperature contributions and relative Gibbs free energies for the adsorbed slabs in Path 4 at 440 K. The DFT total energy E_{elec} and Gibbs free energy G of the experimentally observed species 1 are set as zero points respectively.

	ZPE (eV)	S (eV/K)	$-TS$ (eV)	$\int_0^T C_p(T')dT'$ (eV)	$ZPE+$ $\int_0^T C_p(T')dT'-TS$ (eV)	E_{elec} (eV)	$E_{elec}+ZPE+$ $\int_0^T C_p(T')dT'-$ $TS+\frac{n}{2}\Delta E(H)$ (eV)	G (eV)
1	13.72	5.22×10^{-2}	-22.95	8.62	-0.61	0	-0.61	0
1a	13.77	5.20×10^{-2}	-22.87	8.61	-0.49	-0.77	-1.26	-0.65
1b	13.76	5.14×10^{-2}	-22.63	8.54	-0.33	-1.26	-1.59	-0.98
1b'	13.13	5.17×10^{-2}	-22.76	8.55	-1.07	0.19	-0.78	-0.17
1c	13.16	5.20×10^{-2}	-22.88	8.58	-1.06	0.62	-0.34	0.27
1d	13.18	5.19×10^{-2}	-22.84	8.57	-1.04	0.82	-0.12	0.49
2	12.58	5.14×10^{-2}	-22.62	8.49	-1.53	0.68	-0.75	-0.14

Supplementary Figure 21. Energy diagram of the potential stepwise migration reaction (path 1). (a) Energy diagram with (dark) and without (light) including enthalpic and entropic effects for the stepwise phenyl groups migrations in path 1: dehydrogenations precede phenyl groups migrations. (b) Schematic illustration of the stepwise phenyl group migrations from 1 to 2. (c) Optimized configurations in (a). The experimentally observed species 1 and 2 are denoted with solid lines, while the intermediates 1a, 1b, 1c and 1d are denoted with dashed lines.

Supplementary Figure 22. Energy diagram of the potential stepwise migration reaction (path 2). (a) Energy diagram with (dark) and without (light) including

enthalpic and entropic effects for the stepwise phenyl groups migrations in path 2: dehydrogenations occur during 1-1a and 1a-1b. (b) Schematic illustration of the stepwise phenyl group migrations from 1 to 1. (c) Optimized configurations in (a). The experimentally observed species 1 and 2 are denoted with solid lines, while the intermediates 1a, 1b, 1c and 1d are denoted with dashed lines.

Supplementary Figure 23. Energy diagram of the potential stepwise migration reaction (path 3). (a) Energy diagram with (dark) and without (light) including enthalpic and entropic effects for the stepwise phenyl groups migrations in path 3: dehydrogenations occur during 1b-1c and 1c-1d. (b) Schematic illustration of the stepwise phenyl group migrations from 1 to 1d. (c) Optimized configurations in (a). The experimentally observed species 1 and 2 are denoted with solid lines, while the intermediates 1a, 1b, 1c and 1d are denoted with dashed lines.

Supplementary Figure 24. Energy diagram of the potential stepwise migration reaction (path 4). (a) Energy diagram with (dark) and without (light) including enthalpic and entropic effects for the stepwise phenyl groups migrations in path 4: dehydrogenations occur after two phenyl groups migrations. (b) Schematic illustration of the stepwise phenyl group migrations from 1 to 1d. (c) Optimized configurations in (a). The experimentally observed species 1 and 2 are denoted with solid lines, while the intermediates 1a, 1b, 1c and 1d are denoted with dashed lines.